# Cross-protective antibodies against common endemic respiratory viruses

Madelyn Cabán[1,2,4], Justas V. Rodarte[1,4], Madeleine Bibby[1,4], Matthew D. Gray [1],
Justin J. Taylor [1,2] ✉, Marie Pancera [1] ✉ & Jim Boonyaratanakornkit [1,3] ✉

Respiratory syncytial virus (RSV), human metapneumovirus (HMPV), and human parainfluenza virus types one (HPIV1) and three (HPIV3) can cause severe disease and death in immunocompromised patients, the elderly, and those with underlying lung disease. A protective monoclonal antibody exists for RSV, but clinical use is limited to high-risk infant populations. Hence, therapeutic options for these viruses in vulnerable patient populations are currently limited. Here, we present the discovery, in vitro characterization, and in vivo efficacy testing of two cross-neutralizing monoclonal antibodies, one targeting both HPIV3 and HPIV1 and the other targeting both RSV and HMPV. The 3 × 1 antibody is capable of targeting multiple parainfluenza viruses; the MxR antibody shares features with other previously reported monoclonal antibodies that are capable of neutralizing both RSV and HMPV. We obtained structures using cryo-electron microscopy of these antibodies in complex with their antigens at 3.62 Å resolution for 3 × 1 bound to HPIV3 and at 2.24 Å for MxR bound to RSV, providing a structural basis for in vitro binding and neutralization. Together, a cocktail of 3 × 1 and MxR could have clinical utility in providing broad protection against four of the respiratory viruses that cause significant morbidity and mortality in at-risk individuals.

Respiratory viruses are a major cause of death worldwide, with an estimated 2.7 million attributable deaths in 2015[1]. While a vaccine to prevent RSV infection may be on the horizon[2,3], protective vaccines for HMPV, HPIV3, and HPIV1 are not yet clinically available. Even if protective vaccines existed for these four respiratory viruses, vaccination of highly immunocompromised individuals rarely achieves protective immunity. Additionally, vaccination prior to immune-ablative therapies is often ineffective or wanes quickly, failing to maintain durable protection[4–6]. Together, RSV, HMPV, HPIV1, and HPIV3 represent a serious threat to immunocompromised patients and, prior to the COVID-19 pandemic, were responsible for the majority of viral lower respiratory infections in hematopoietic stem cell transplant recipients[7,8]. In adults with other risk factors, the burden of disease from HMPV and the parainfluenza viruses is also comparable to RSV[9,10]. Further, with the exception of rhinoviruses, RSV, HMPV, and the

parainfluenza viruses also collectively account for most of the respiratory viruses identified in hospitalized adults prior to 2020[11,12].

Although mitigation strategies during the COVID-19 pandemic such as masking, social distancing, and shut-downs led to declines in cases of other respiratory viruses during the 2020–2021 cold and flu seasons, cases of RSV, HMPV, and HPIVs are beginning to surge again, and are expected to return to pre-pandemic levels of circulation in the next few years[9,10]. In fact, models project large future outbreaks of non-SARS-CoV-2 respiratory viruses due to an increase in the size of the susceptible population following a period of reduced spread[13]. Additionally, since endemic respiratory viruses tend to circulate seasonally, co-infections with more than one respiratory virus can occur and have been associated with worse outcomes in vulnerable populations[14–16].

The administration of neutralizing monoclonal antibodies (mAbs) provides an effective alternative to vaccination to protect against viral

[1]Vaccine and Infectious Disease Division, Fred Hutchinson Cancer Center, Seattle, WA, USA. [2]Department of Immunology & Department of Global Health, University of Washington, Seattle, WA, USA. [3]Department of Medicine, University of Washington, Seattle, WA, USA. [4]These authors contributed equally: Madelyn Cabán, Justas V. Rodarte, Madeleine Bibby. ✉e-mail: jtaylor3@fredhutch.org; mpancera@fredhutch.org; jboonyar@fredhutch.org

infections. Although the anti-RSV mAb palivizumab received FDA approval in 1998 as prophylaxis in high-risk infants[17], it remains relatively unused in older immunocompromised children or adults. Since the approval of palivizumab, even more potent mAbs against RSV have progressed through clinical trials, with the primary goal of replacing palivizumab as the standard of care for prophylaxis in high-risk infants. This focus has been driven in part because RSV causes up to 80% of bronchiolitis in infants[18,19]. However, in immunocompromised adults, the respiratory virus landscape is much more heterogeneous[7,8]. Therefore, an effective strategy to reduce the overall burden of the broad range of lower respiratory tract infections in at-risk adults and immunocompromised patients must rely on targeting multiple viruses simultaneously, rather than a single virus. Despite advances in research on RSV prevention, the role of passive immunization for other respiratory viruses remains poorly defined and no mAbs are currently available in the clinic that can prevent HMPV, HPIV1, or HPIV3 infection.

To efficiently achieve broader protection against these viruses, we sought to identify cross-neutralizing mAbs that could target more than one virus at a time. RSV, HMPV, HPIV3, and HPIV1 all produce class I fusion (F) proteins which are essential surface glycoproteins specialized to mediate fusion between viral and host cell membranes during viral entry. HPIV1 and HPIV3 belong to the same *Respirovirus* genus, and their F sequences share 65% amino acid sequence homology. RSV and HMPV belong to the same *Pneumoviridae* family, and their F sequences share ~54% homology. The F proteins transition between a metastable prefusion (preF) conformation and a stable postfusion (postF) conformation[20,21]. Since preF is the major conformation of infectious virions, antibodies to preF tend to be the most potent at neutralizing virus[22-25]. Similar to RSV, the HPIV3 and HPIV1 F proteins in

the prefusion conformation elicit higher neutralizing antibody titers compared to the postfusion conformation[26]. In a previous study, we used the stabilized HPIV3 preF protein to identify and characterize several neutralizing antibodies against HPIV3[27]. For HMPV, even though antibodies targeting the post-fusion conformation also contribute to neutralizing antibody titers[28], HMPV postfusion F does not elicit cross-neutralizing antibodies to RSV[29]. In the present study, we leveraged the homology between the related F proteins and focused on their preF conformations to identify and clone two potent cross-neutralizing mAbs, 3×1 and MxR, from human memory B cells. 3×1 cross-neutralizes both HPIV3 and HPIV1, while MxR effectively cross-neutralizes both HMPV and RSV. Together, 3×1 and MxR comprise an antibody cocktail with the ability to achieve simultaneous protection against multiple viruses which could be beneficial to at-risk populations who are at a significant immunological disadvantage when infected with respiratory viruses.

## Results

### Identification of an HPIV3 and HPIV1 cross-neutralizing antibody

Since virtually all humans have been exposed to RSV, HMPV, HPIV3, and HPIV1[30], it was not necessary to pre-screen donors for seropositivity. Because there is a lack of mAbs currently under development against the parainfluenza viruses, we first focused our efforts on screening for HPIV3/HPIV1 cross-neutralizing B cells. Since we were unable to produce the HPIV1 F protein in the preF conformation, to conduct this screen we used the F protein of HPIV3 alone by leveraging a bait-and-switch strategy (Fig. 1a, b). Here, HPIV3-binding B cells were isolated by incubating 200 million human splenocytes with tetramers of HPIV3 preF conjugated to allophycocyanin (APC) and tetramers of HPIV3 postF conjugated to APC/Dylight755 followed by magnetic

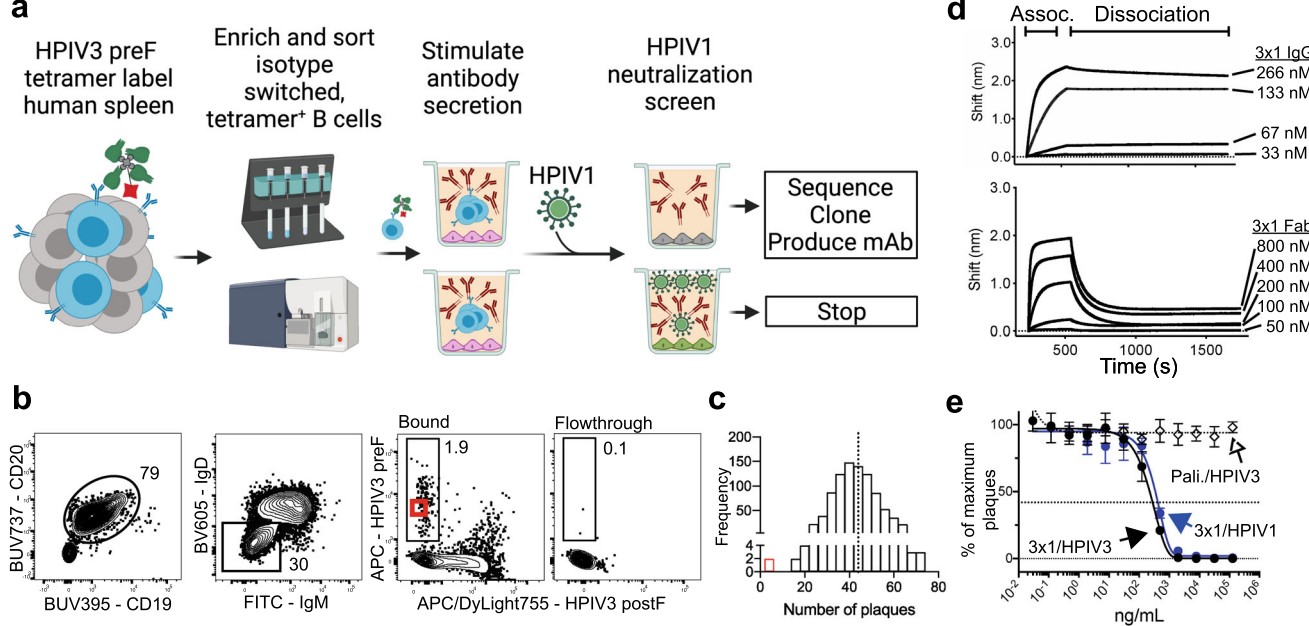

**Fig. 1 | Identification of an HPIV3/HPIV1 cross-neutralizing monoclonal antibody. a** Bait-and-switch approach using a single antigen to identify B cells that cross-neutralize another related virus. HPIV3-binding B cells from human spleen were labeled with APC-conjugated tetramers of HPIV3 preF. **b** Flow cytometry plot of HPIV3 preF-binding B cells after gating for live, CD3⁻CD14⁻CD16⁻CD19⁺CD20⁺ (B cells), IgD⁻/IgM⁻ (isotype-switched), and APC/Dylight755⁻ HPIV3 postF⁻ (to exclude cells binding to HPIV3 postF, APC, or streptavidin). The bound fraction contains cells magnetically enriched cells using APC-specific microbeads. Numbers on plots are percentages of total cells in the gate. The red box indicates the B cell from which 3×1 mAb was derived. **c** Frequency distribution of HPIV1 plaques per well from the neutralization screen. The dotted line indicates the mean number of plaques in

negative control wells containing the virus in the absence of antibodies. The red bar includes data from the well that contained the 3×1-producing B cell. **d** Binding kinetics of 3×1 IgG (top) and Fab (bottom) to HPIV3 preF were measured by bio-layer interferometry (BLI) to determine apparent affinities ($K_D$). Association with 3×1 to HPIV3 preF-loaded probe was measured for 300 s followed by dissociation for 1200 s. Measurements are normalized against an isotype control antibody. **e** Vero cells were infected with HPIV3 or HPIV1 in the presence of serial dilutions of palivizumab or 3×1. The dotted midline indicates PRNT₆₀. Data points are the average ± SD from three independent experiments. Panel (**a**) created with BioRender.com.

enrichment with anti-APC microbeads. Nine-hundred B cells that bound HPIV3 preF tetramers but not postF tetramers were individually sorted into wells containing CD40L/IL2/IL21-producing 3T3 feeder cells and then incubated for 13 days to stimulate antibody secretion into culture supernatants. To identify wells containing candidate B cells expressing cross-neutralizing antibodies, culture supernatants from the individually sorted HPIV3-binding B cells were mixed with live HPIV1 virus and screened for their ability to reduce plaque formation (Fig. 1a, c). Two out of the 900 HPIV3-binding B cells produced antibodies that could neutralize HPIV1 (Fig. 1c). From these cells, the expressed heavy (VH3-23) and light (Vλ3-19) chain genes were sequenced and cloned successfully to produce a mAb with cross-neutralizing capability against HPIV3/HPIV1 which we named $3 \times 1$ (Fig. 1b, c). $3 \times 1$ was isolated from a B cell expressing the IgA isotype. Since palivizumab and most other mAbs being developed against respiratory viruses utilize an IgG1 constant region, $3 \times 1$ was also cloned and produced for further study as an IgG1.

Although we did not have the F protein of HPIV1 stabilized in the preF conformation, we did estimate the apparent binding affinity between $3 \times 1$ and the F protein of HPIV3 in the preF conformation. As an IgG, $3 \times 1$ bound tightly to the preF protein of HPIV3 ($K_D < 10^{-12}$ M) (Fig. 1d). The binding affinity of $3 \times 1$ Fab was lower ($K_D = 1.9 \times 10^{-7}$ M), due to rapid dissociation, indicating that simultaneous binding by both Fabs is likely required to maintain binding (Fig. 1d). The neutralizing potency of $3 \times 1$ was determined by a 60% plaque reduction neutralization test ($PRNT_{60}$) using live virus to infect Vero cells. $3 \times 1$ had a similarly high neutralization potency against both HPIV1 and HPIV3, with a $PRNT_{60}$ of 352 and 242 ng/mL, respectively (Fig. 1e). Further investigation revealed that $3 \times 1$ blocked cell-to-cell spread and syncytia formation by HPIV3 in cell culture (Fig. S1). These results suggest that the epitope of $3 \times 1$ might be functionally conserved between HPIV3 and HPIV1.

## Cryo-EM structure of $3 \times 1$ Fab in complex with HPIV3 preF

Since the antigenic landscape of HPIV3 preF is not well characterized, we first performed cross-competition binding experiments to gauge the antigenic sites on HPIV3 preF allowing for neutralization. The epitope for the cross-neutralizing mAb $3 \times 1$ did not appear to overlap with the epitopes of mAbs that we had previously isolated which bind to the apex of preF and neutralized HPIV3 (Fig. 2a)[27]. However, the epitope of $3 \times 1$ did appear to overlap with the epitope of a previously described antibody PI3-A12, which binds to an antigenic site we named site X[27]. To determine how $3 \times 1$ interacts with site X, we used cryo-EM. Many HPIV3 preF trimer particles had <3 Fabs bound, despite the Fab being in molar excess of trimer during sample preparation (Fig. S3a). We obtained a structure of 1 Fab in complex with HPIV3 resolved to 3.62 Å (Fig. S2a and Table S1). We also obtained a structure of 3 Fabs bound to HPIV3; this map was limited to 4.3 Å resolution (Figs. S2a and S3a). We noticed no significant variation between the bound preF protomer and unbound preF protomer (RMSD = 0.721 over 360 Cα) in the C1 structure. Consequently, we used the higher resolution structure for model building. $3 \times 1$ binds domain III of HPIV3 preF, protruding perpendicularly and binding only one protomer, with no additional contacts to other regions (Fig. 2b). Only four of the six CDRs are involved in binding, with CDRH1 and CRDL2 being too short to contact the glycoprotein surface (Fig. 2c, d). $3 \times 1$ binds site X with a total buried surface area (BSA) of ~812 Å², of which the VH contributes ~612 Å² and the VL contributes ~200 Å² (Fig. 2b, c). Comparison with the PI3-A12:HPIV3 structure indicated that $3 \times 1$ binds the same site but is rotated relative to HPIV3 preF (Fig. S3b). While $3 \times 1$ neutralizes both HPIV3 and HPIV1, PI3-A12 neutralizes only HPIV3[27], and $3 \times 1$ shares little CDR sequence similarity with PI3-A12 (Fig. S3c). Due to the lack of a high-resolution structure for the PI3-A12:HPIV3 complex, we could not determine if the $3 \times 1$ LC overlaps with the PI3-A12 LC or HC. $3 \times 1$ and PI3-A12 may therefore bind distinct epitopes within the same site of

HPIV3 preF, as they display significant sequence variation at key residues in $3 \times 1$ that facilitate binding (Fig. S3c).

Further analysis of the local resolution of our map indicated the binding site had a higher resolution of ~3.0 Å compared to the overall resolution of 3.62 Å (Fig. S1a). The $3 \times 1$ VH and VL together specifically bind the cleft between the HRA helix and a sheet-turn-sheet motif, a short contiguous region of HPIV3 (Figs. S4a–d and S5a, b). This region is crucial for the preF to postF rearrangement, with both the HRA helix and sheet-turn-sheet motif displaying >9 Å of movement during rearrangement[26]. The HRA may also play a role in HPIV1 neutralization since $3 \times 1$ likely interacts with the HRA on HPIV1 (Fig. S4e). Due to the significant motion required of this site during fusion, and its potency as a neutralizing epitope for RSV preF[31], $3 \times 1$ binding at this location presents a strong structural basis for the high neutralizing potency of $3 \times 1$. The CDRH2 and CDRH3 of $3 \times 1$ form several protrusions into grooves on the surface of HPIV3 preF using non-polar residues (Fig. 2d, e). His52A_{HC} and Phe56_{HC} (CDRH2) along with Leu100B_{HC} and Leu100F_{HC} (CDRH3) account for ~55% of the total VH BSA (Fig. 2c). Most other contacting residues within the VH are polar residues which form contacts with generally <50 Å² of BSA. The VL residues of $3 \times 1$ form contacts with HPIV3 preF using polar functional groups and the CDR loop backbone, with CDRL1 encompassing a large protrusion on HPIV3 preF (Fig. 2c, f). This CDRL1 extension is supported by Arg91_{LC}, which contacts Tyr31_{LC} and Leu100F_{HC}, with Arg91_{LC} forming a bond with Asp143 of HPIV3 preF, which is conserved in HPIV1 (Figs. S5a–c and S6a). We also note a poorly resolved feature in our $3 \times 1$:HPIV3 map, which may be the C-terminus of the F2 protein following furin cleavage (Fig. S7a, b). While there is not sufficient density to build this region, its proximity to the $3 \times 1$ binding site could indicate an additional binding epitope.

## Identification of a potent RSV and HMPV cross-neutralizing antibody

Since RSV and HMPV also contribute significantly to disease in vulnerable patients, we next sought to identify potential HMPV/RSV cross-neutralizing mAbs that could be combined with $3 \times 1$ to create a potent cocktail with expanded breadth. Although many mAbs targeting RSV and HMPV are already in development, we used our approach leveraging fluorescent tetrameric probes to identify unique B cells able to bind recombinant F proteins from RSV and HMPV in the preF, but not in the postF, conformation. RSV preF conjugated to APC, HMPV preF conjugated to R-phycoerythrin (PE), RSV postF conjugated to APC/DyLight755, and HMPV postF conjugated to PE/DyLight650 were mixed with 200 million peripheral blood mononuclear cells (PBMCs) prior to magnetic enrichment with anti-PE and anti-APC microbeads. We then sorted single isotype-switched B cells binding to the F proteins of RSV and HMPV in the preF conformation but not in the postF conformation, followed by single-cell sequencing and cloning of their B cell receptors (Fig. 3a). Using this method, the heavy (VH3-21) and light (Vλ1-40) chain alleles of a B cell capable of binding both RSV and HMPV were sequenced and cloned as a mAb which we named MxR (Fig. 3b). This B cell expressed the IgG isotype. Like $3 \times 1$, MxR was cloned and produced as an IgG1 for further study.

We compared the apparent binding affinity of MxR to the RSV-specific monoclonal antibody D25, which is being developed under the name nirsevimab for prophylaxis of RSV in infants[32,33]. RSV has two antigenically distinct subtypes, A and B, which share 91% amino acid sequence homology within the F protein. MxR bound irreversibly with high apparent affinity to the preF proteins of both RSV subtypes A and B ($K_D < 10^{-12}$ M each), even when the dissociation time was extended to 1200 s (Figs. 3c and S8a, d). The previously reported cross-neutralizing monoclonal antibody MPE8[34] also exhibited a high apparent affinity for both RSV subtypes A and B (Figs. 3c and S8b, e). D25 also bound strongly to the preF protein from RSV subtype B, but with ~1500-fold lower apparent affinity ($K_D = 1.5 \times 10^{-9}$ M) compared to its binding to

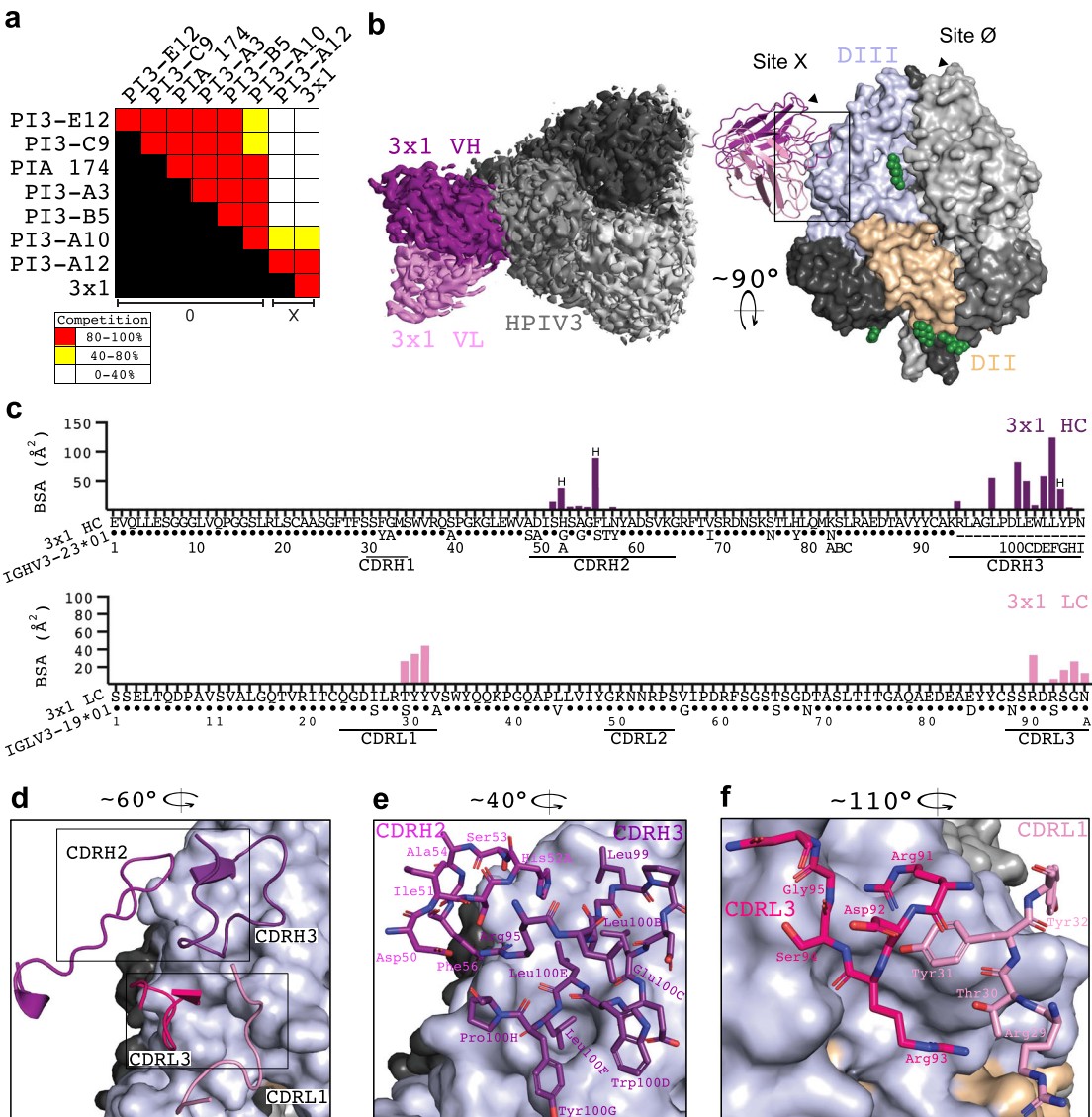

**Fig. 2 | Analysis of the HPIV3 preF epitope bound by 3 × 1. a** BLI measurement of the ability of the mAb listed on the left side of the chart to prevent binding of the mAb listed on the top, expressed as the percent drop in maximum signal compared to the maximum signal in the absence of competing mAb. **b** Cryo-EM structure of 3 × 1 Fab in complex with HPIV3 preF. Left, a top-down view of the 3 × 1:HPIV3 map is shown with one Fab ($V_H$ shown in purple, $V_L$ in pink) and HPIV3 preF in shades of gray. Right, a surface representation of the complex is shown rotated 90° with domain III in light blue and domain II in yellow on one protomer. Glycans are shown as green spheres. **c** BSA plots for each Fab residue that interacts with the preF protomer, atop a sequence alignment to the germline $V_H$ and $V_L$ for 3 × 1. **d** Detailed view of the 3 × 1 binding site with only the interacting CDRs shown in cartoon, rotated 60° from panel (**b**). Boxes show the locations of panels (**e**) and (**f**). **e** Zoomed-in view of the CDRH3 binding site rotated 40° from panel (**b**). **f** Zoomed-in view of the CDRL3 binding site rotated 110° from panel (**b**). **e, f** CDR residues that make no contact with HPIV3 preF have been hidden from view for clarity.

subtype A (Figs. 3c and S8c, f). MxR also could bind to the preF protein of HMPV (Figs. 3c and S8g), which was expected given the deliberate selection of B cells binding both RSV and HMPV during the sort. Compared to MPE8, the apparent binding affinity of MxR for HMPV was approximately 1.6-fold stronger (Figs. 3c and S8g, h).

Since both subtypes of RSV circulate globally, it was important to assess the neutralization potency ($PRNT_{60}$) of MxR for subtypes A and B. We also compared the neutralizing potency of MxR with the RSV-specific monoclonal antibody palivizumab, which is currently approved for RSV prophylaxis in high-risk infants. We found that MxR neutralized RSV subtype A with at least 6-fold greater potency as compared to palivizumab (Figs. 3d and S9a). In contrast to palivizumab, which has similar potency against both subtypes[35], MxR was also highly potent against subtype B, with a 12-fold greater potency (Figs. 3d and S9b). Although D25 had greater neutralizing potency against RSV-A (Fig. 3d), D25 reportedly has weaker potency against

some strains of RSV-B and does not neutralize HMPV[36–38]. MxR and MPE8 had similar potency in neutralizing both RSV subtypes A and B. However, MxR had at least 5-fold greater potency against HMPV, with a $PRNT_{60}$ of 148 ng/mL compared to 838 ng/mL for MPE8 (Figs. 3d and S8c). Based on the concentration of antibody needed to neutralize 60% of live virus, the potency of MxR against HMPV exceeded the relative potency of palivizumab against RSV (Fig. 3d).

**Cryo-EM structure of MxR Fab in complex with RSV preF**
To better understand the basis for the cross-neutralization observed with MxR, we obtained a cryo-EM structure of three MxR Fabs bound to RSV preF to 2.24 Å resolution (Figs. S2b, S10, and Table S1). MxR binds primarily to antigenic site III on RSV with an equatorial arrangement of Fabs around RSV preF. Antigenic site III is a quaternary epitope at the junction of domains I and III of one F protomer and domain II of the clockwise adjacent F protomer (referred to as II′)

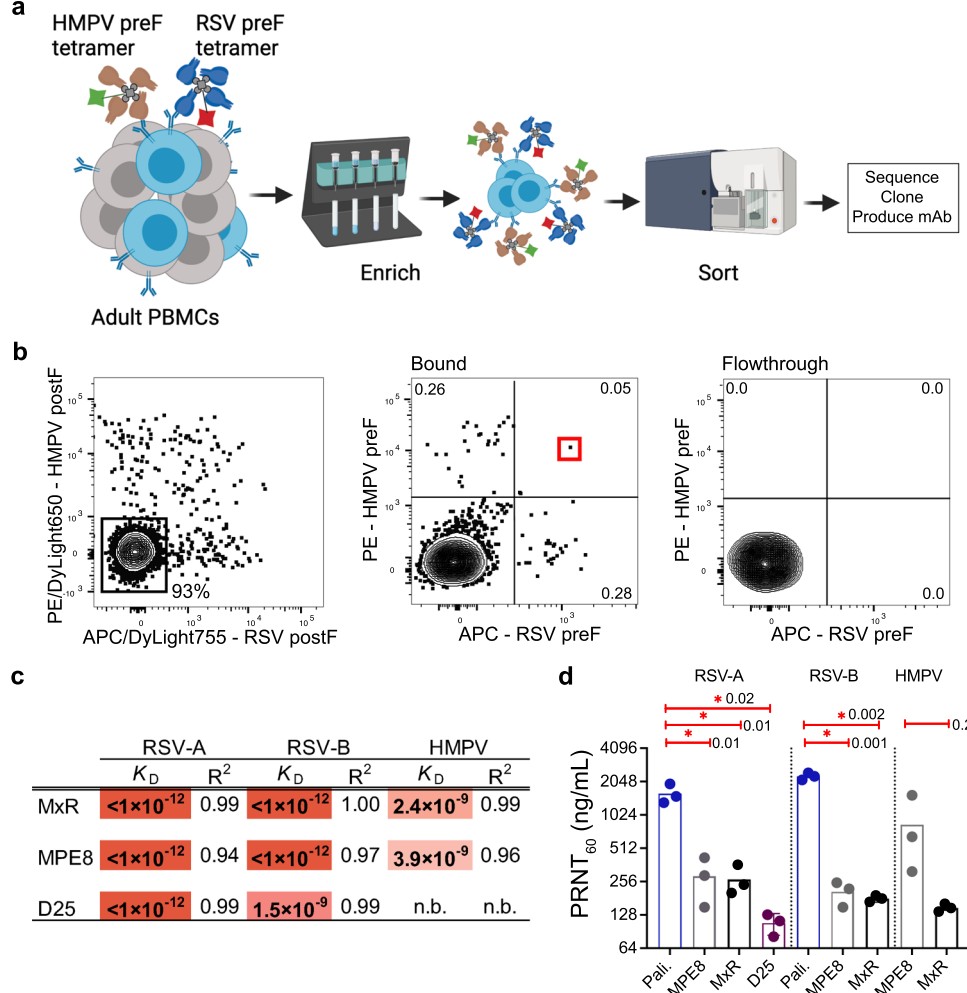

**Fig. 3 | Identification and analysis of HMPV/RSV cross-neutralizing monoclonal antibodies. a** RSV- and HMPV-binding B cells from human blood were labeled with APC-conjugated streptavidin tetramers of biotinylated RSV prefusion protein (preF) and PE-conjugated streptavidin tetramers of biotinylated HMPV preF. **b** Flow cytometry plot of RSV and HMPV preF-binding B cells after gating for live, CD3⁻CD14⁻CD16⁻ CD19⁺CD20⁺ (B cells), IgD⁻/IgM⁻ (isotype-switched), and APC/Dylight755⁻ HMPV postF⁻ and PE/Dylight650⁻RSV postF⁻ (to exclude cells binding to RSV/HMPV postF, APC, PE, or streptavidin). The bound fraction contains magnetically enriched cells using APC- and PE-specific microbeads. Numbers on plots are percentages of total cells in the gate. The red box indicates the B cell from which MxR mAb is derived. **c** Apparent affinity ($K_D$) of MxR, MPE8, and D25 measured by BLI. **d** Vero cells were infected with RSV-A, RSV-B, or HMPV in the presence of serial dilutions of palivizumab (Pali.), D25, MPE8, or MxR. Data points represent the 60% plaque reduction neutralization titer and are from three independent experiments. The asterisks indicate a $p < 0.05$ using a two-tailed $t$-test with Welch's correction. Numbers next to asterisks indicate the exact $p$-value. Panel (**a**) created with BioRender.com.

(Fig. 4a). MxR binds with a total BSA of ~1094 Å², with VH contributing ~694 Å² and the VL contributing ~400 Å², of which ~298 Å² contacts with the main F protomer and ~102 Å² contacts with the adjacent F protomer. Our structure revealed numerous water molecules spaced throughout the binding site that potentially mediate interactions between the Fab and preF protomer (Fig. S11). These were omitted from Fig. 4 for clarity. The mode of binding is almost identical to the previously reported cross-neutralizing monoclonal antibody MPE8[34] and the infant monoclonal antibody ADI-19425[39], which are derived from the same germline heavy (IGHV3-21*01) and light chain (IGLV1-40*01) alleles. Comparison of the per-residue BSA of RSV preF in complex with MxR, MPE8, and ADI-19425 revealed common binding regions that share high sequence homology with HMPV (Fig. S6b). The sequence and structure of both CDR1s and CDR2s are nearly identical across all three antibodies, with MxR overall having more mutations from germline than the other two (Fig. 4b, c).

The CDRH3 regions of MxR, MPE8, and ADI-19425 have a greater degree of sequence and structural variation, with almost no conserved residues, despite all binding near the DI/DII'/DIII interface within

antigenic site III. There is a small cleft at this interface, into which the CDRH3 loops extend to varying degrees (Fig. 4c, d). MPE8 CDRH3 is the longest, while MxR is the shortest and ADI-19425 is of intermediate length. Notably, ADI-19425 uses a disulfide bond to stabilize this loop, whereas MPE8 and MxR lack this bond (Fig. 4d)[34,39]. The rigid geometry imposed by the disulfide bond may impair ADI-19425's ability to bind HMPV. The necessity of the MPE8 CDRH3 in forming the correct loop geometry to bind within the cleft of antigenic site III may thus be a structural basis behind MPE8's reduced neutralization of HMPV compared to MxR (Figs. 4c, d and 3d). Due to a relatively shorter CDRH3 than MPE8, MxR binds in this cleft without contacting DII', and therefore does not require an extended loop to facilitate binding (Figs. 4d and S6b). Only the CDRL2 of MxR contacts DII' (Fig. 4c). When the CDRs are superimposed on the HMPV protomer, we find that the general binding site is highly similar, comprising similar antigenic site contributions and residue identities (Fig. S12a, b). However, the available binding cleft is narrower and shorter, which may sterically hinder ADI-19425 and MPE8 CDRH3 loops (Fig. S12c). The CDRL3s also show some variation in interacting residues (Fig. 4e). Both MPE8 and MxR

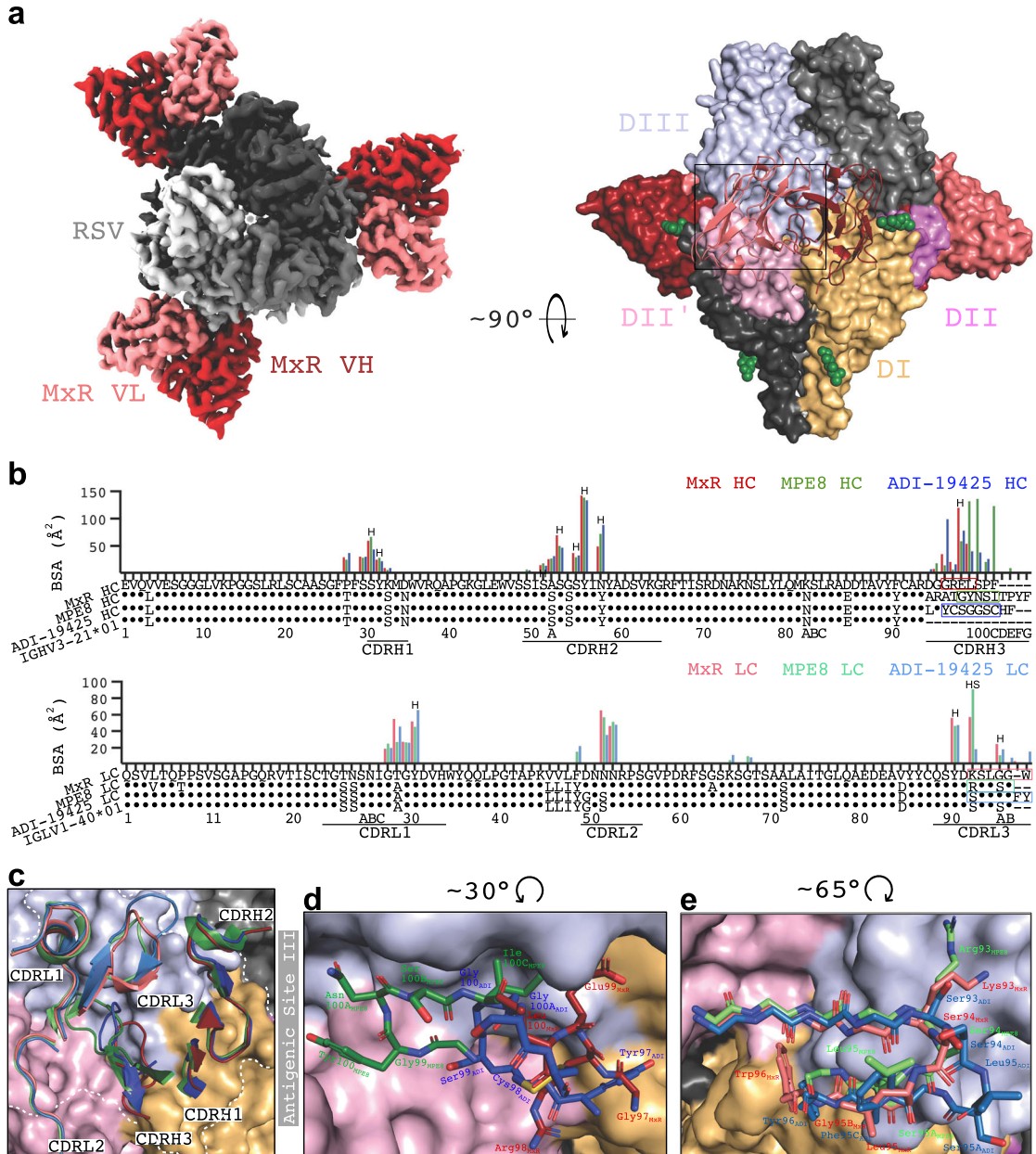

**Fig. 4 | Cryo-EM structure of MxR Fab in complex with RSV preF and comparison with MPE8 and ADI-19425. a** Left, a top-down view of the DeepEMhanced MxR:RSV cryo-EM map is shown, with MxR $V_H$ in dark red, MxR $V_L$ in light red, and RSV preF in shades of gray. Only the Fv domain of the Fab is shown. Right, a surface representation of the structure is shown rotated 90°. One preF protomer is colored per its structural domains, and one MxR Fv is shown in a cartoon outline. The DII domain of the clockwise-adjacent protomer is in light pink, and designated DII' to differentiate it from DII of the other protomer, colored in violet. Glycans are shown as green spheres. (**b**) BSA plots for MxR, MPE8 and ADI-19425 residues which interact with RSV preF, atop a sequence alignment of the germline V-genes. The letter H indicates hydrogen bonds. The letter S indicates salt bridges. Residues in colored boxes correspond to the residues shown in panels (**d** and **e**). **c** Detailed view of the binding site on RSV preF with the CDRs of MxR, MPE8, and ADI-19425 superimposed and shown in the cartoon. MPE8, in greens, and ADI-19425, in blues, are shown in transparency. Antigenic site III is delineated by the white outline. **d** Zoomed-in view of the CDRH3 binding site rotated 30° counterclockwise from panel (**c**). **e** Zoomed-in view of the CDRL3 binding site, rotated 65° clockwise from panel (**c**). The first four residues are shown as mainchain only to increase clarity and illustrate the similarity of CDRL3 between antibodies.

share a basic residue at position $93_{LC}$ which is not shared with ADI-19425, and all three antibodies exhibit slightly different loop arrangements in residues $94–96_{LC}$ (Fig. 4e). However, the binding modes of the CDRL3s appear similar, without the unique features seen in the CDRH3s.

## In vivo protection against viral infection
We next investigated whether the in vitro binding and neutralization data would translate into in vivo protection in an animal challenge model. Although the human parainfluenza viruses do not replicate in

mice, upper and lower respiratory tract replication can be demonstrated in both hamsters and cotton rats[30,40]. For RSV, comparable viral titers have been reported in the lungs of hamsters and cotton rats[41]. Although some studies have observed higher titers of HMPV in the lungs of cotton rats compared to hamsters[42,43], this could be related to differences in the viral strain used, the size of the inoculum, and the timing of lung sampling. The hamster model has been used extensively to evaluate vaccine candidates for parainfluenza viruses, RSV, and HMPV[44–52]. Since all the viruses in this study could replicate in hamsters[44,45,51,53–57], we performed preclinical testing of MxR and 3 × 1 in

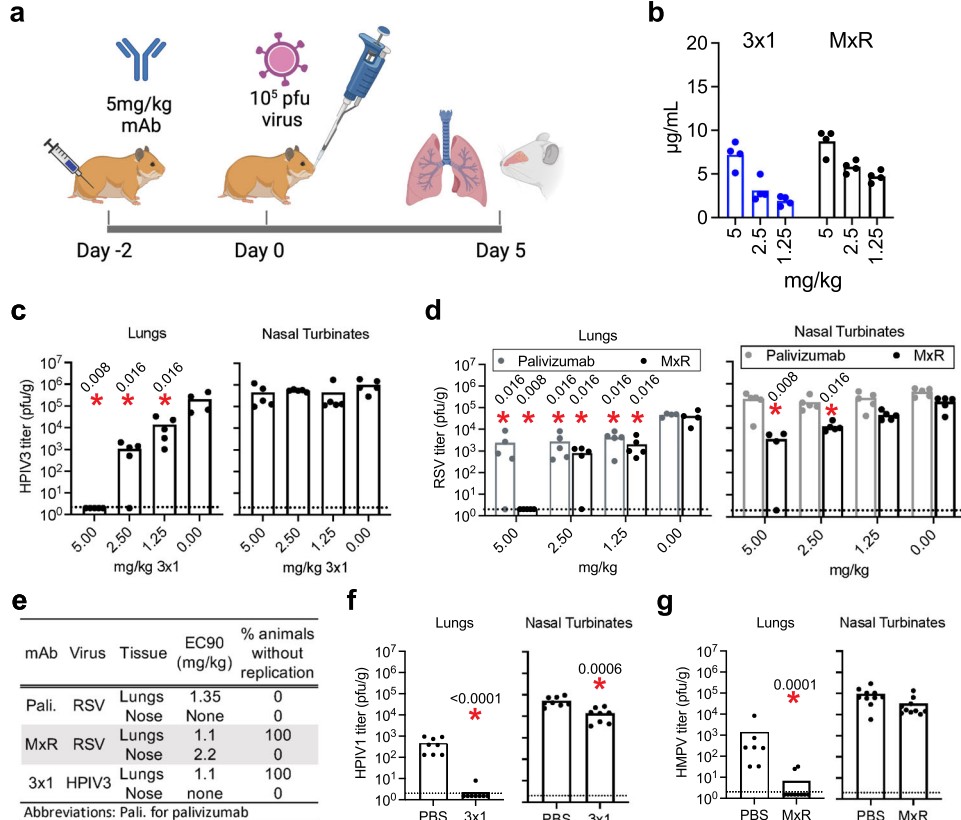

**Fig. 5 | Efficacy of prophylactic administration of cross-neutralizing monoclonal antibodies 3 × 1 and MxR in vivo. a** Schematic of experiments in which hamsters were injected intramuscularly with 3 × 1 or MxR two days prior to intranasal challenge with $10^5$ pfu of virus. **b** Serum concentrations of 3 × 1 and MxR two days after administration of mAb at 5, 2.5, and 1.25 mg/kg were measured by ELISA ($n = 4$ animals/dose). Dose–response of 3 × 1 (**c**) and MxR (**d**) on HPIV3 and RSV replication, respectively ($n = 5$ experimental animals/dose/virus, $n = 5$ control animal nasal turbinates/virus, and $n = 4$ control animal lungs/virus). **e** 90% effective concentration ($EC_{90}$) of palivizumab against RSV, MxR against RSV, and 3 × 1 against HPIV3 based on dose–response experiments. **f** HPIV1 and (**g**) HMPV replication

after injection with 3 × 1 or MxR at 5 mg/kg, respectively ($n = 8$ animals/group over two independent experiments for HPIV1; and $n = 10$ animal nasal turbinates/group, $n = 10$ experimental animal lungs, and $n = 7$ control animal lungs over two independent experiments for HMPV). Viral titers were measured by plaque assay in lung and nasal homogenates from individual hamsters at 5 days post-infection. Dashed lines indicate the limit of detection. Bars represent the mean, and asterisks indicate two-sided $p < 0.05$ by Mann–Whitney test compared to control hamsters injected with 1× DPBS. Numbers next to asterisks indicate the exact $p$-value. Panel (**a**) created with BioRender.com.

the Golden Syrian hamster model. We performed intramuscular injections in hamsters on day −2, infected hamsters intranasally on day 0, and harvested lungs and nasal turbinates on day 5 post-infection to assess the efficacy of monoclonal antibodies as prophylaxis against infection (Fig. 5a). This dosing, route, and timing of administration are similar to those used in cotton rat models of RSV infection[41,58,59]. We performed dose-finding experiments with 3 × 1 against HPIV3, and MxR against RSV, using palivizumab as a benchmark. Two days after intramuscular injection of the 5 mg/kg dose in hamsters, we observed mAb serum concentrations of 5.1–9.7 μg/mL (Fig. 5b). At a dose of 5 mg/kg, MxR and 3 × 1 fully suppressed viral replication of HPIV3 and RSV, respectively, in the lungs (Fig. 5c, d). In contrast, palivizumab at 5 mg/kg did not completely suppress viral replication in the lungs, even though palivizumab and MxR had a similar $EC_{90}$ against RSV (Fig. 5d, e). This is consistent with data from the cotton rat model in which breakthrough infection with palivizumab at 5 mg/kg was also observed[60]. Prophylactic administration of 3 × 1 at 5 mg/kg had little impact on replication in the nasal turbinates (Fig. 5c). However, prophylactic administration of MxR at 5 mg/kg reduced RSV replication in nasal turbinates by over 47-fold (Fig. 5d). This is in contrast to palivizumab, which had no effect on RSV replication in the nasal turbinates. Since the 5 mg/kg dose suppressed viral replication of RSV and HPIV3, we also tested this dose for HPIV1 and HMPV. HPIV1 replication was completely blocked in the lungs of all but one animal and was

significantly reduced in the nasal turbinates of hamsters that received 3 × 1 prophylaxis (Fig. 5f). HMPV replication was significantly reduced by over 206-fold in the lungs of hamsters that received MxR prophylaxis (Fig. 5g).

If administered together as a cocktail, MxR and 3 × 1 could provide broad protection against HMPV, RSV, HPIV3, and HPIV1. This is clinically relevant because all four viruses together account for the majority of serious respiratory viral infections in hematopoietic stem cell transplant recipients. Compared to the co-administration of four mAbs (one per virus), using two mAbs to target four viruses allows for the administration of a higher dose of each mAb, which could maximize efficacy while minimizing toxicity. Administration of a cocktail could also be useful in the setting of co-infections with multiple respiratory viruses since co-infections can be associated with poorer outcomes in immunocompromised patients[16]. We, therefore, developed an HPIV3/RSV co-infection model in hamsters to assess efficacy when MxR and 3 × 1 are co-administered together. Since the plaque assay to determine viral titers could not distinguish between HPIV3 vs. RSV, we developed custom TaqMan probes to quantify the individual viral loads by real-time PCR. First, we compared the levels of HPIV3 and RSV detected in the lungs of animals infected with one or both of these viruses. Similar to data from human studies suggesting that co-infections between RSV and other viruses do not have an impact on RSV titers[61], we did not observe any decrease in viral replication in the lungs of animals

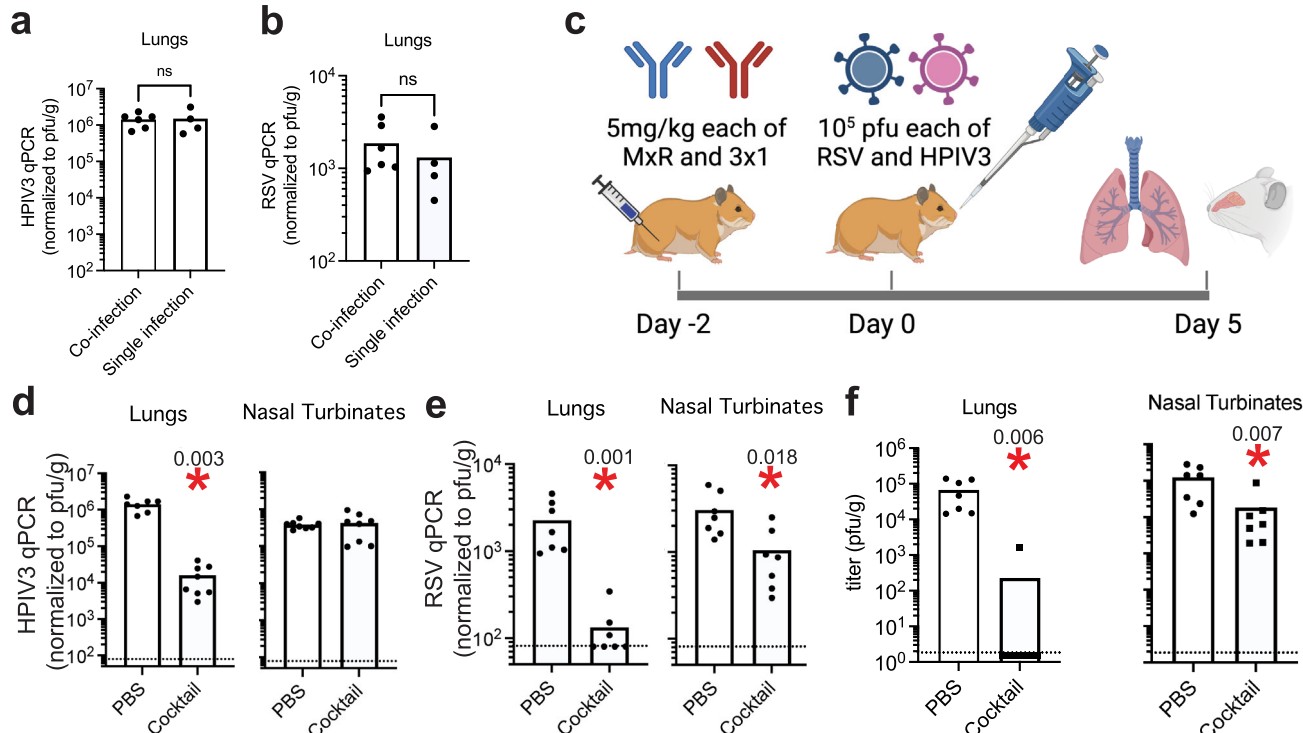

**Fig. 6 | Efficacy of prophylactic administration of a cocktail of 3 × 1 and MxR against HPIV3 and RSV co-infection in vivo.** The specific viral load of HPIV3 (**a**) and RSV (**b**) in lung homogenates of hamsters co-infected with $10^5$ pfu each of RSV and HPIV3 was compared to hamsters infected with $10^5$ pfu of a single virus by real-time PCR: $n = 4$ animals/single infection and $n = 6$ for the co-infection. **c** Schematic of experiments in which hamsters were injected intramuscularly with 5 mg/kg each of 3 × 1 and MxR as a cocktail two days prior to intranasal challenge with $10^5$ pfu each of HPIV3 and RSV. The specific viral load of HPIV3 (**d**) and RSV (**e**) in lung and nasal tissue homogenates was determined by real-time PCR using HPIV3- and RSV-specific primers, respectively. For HPIV3: $n = 8$ animal nasal turbinates/group, $n = 8$ experimental animal lungs, and $n = 7$ control animal lungs over two independent experiments. For RSV: $n = 7$ animals/group over two independent experiments. **f** Overall viral titer by plaque assay in lung and nasal homogenates at day 5 post-infection ($n = 7$ animals/group over two independent experiments). Dashed lines indicate the limit of detection, bars represent the mean, and asterisks denote two-sided $p < 0.05$ by the Mann–Whitney test compared to control hamsters. Numbers next to asterisks indicate the exact $p$-value. n.s. for non-significant. Panel (**c**) created with BioRender.com.

simultaneously inoculated with equal amounts of RSV and HPIV3, compared to animals inoculated with a single virus (Fig. 6a, b).

We next injected hamsters intramuscularly with a cocktail of MxR and 3 × 1 on day −2, co-infected hamsters with HPIV3 and RSV on day 0, and harvested lungs and nasal turbinates on day 5 post-infection (Fig. 6c). The cocktail of antibodies did not have an impact on HPIV3 replication in the nasal turbinates but substantially reduced the viral load in the lungs by over 88-fold (Fig. 6d). The cocktail of antibodies specifically reduced RSV viral load in the lungs and nasal turbinates by over 17-fold and 2.9-fold, respectively, and RSV was below the limit of detection in the lungs of four out of seven hamsters (Fig. 6e). Using a plaque assay, the cocktail of antibodies significantly reduced combined viral replication of HPIV3 and RSV in the lungs to undetectable levels in 6 out of 7 animals and by over 6-fold in nasal turbinates (Fig. 6f).

## Discussion

We have isolated two anti-viral cross-neutralizing monoclonal antibodies: 3 × 1 which targets HPIV3 and HPIV1; and MxR which targets RSV and HMPV. Combined, these two antibodies could provide simultaneous and broad protection against four of the major respiratory viruses that afflict hematopoietic stem cell transplant patients and other vulnerable populations. For this, we developed a bait-and-switch strategy based on the rationale that B cells capable of binding to one virus, while neutralizing another virus, are more likely to cross-neutralize both viruses. This strategy led to the discovery of 3 × 1, a monoclonal antibody that neutralizes multiple parainfluenza viruses. The bait-and-switch strategy could also be a generally useful approach

for identifying cross-neutralizing antibodies against other pathogens in a situation where not all antigens are known or available. We also used magnetic enrichment and cell sorting to isolate rare B cells that could bind specifically to recombinant fusion proteins in the preF but not the postF conformation. Further, we leveraged feeder cells expressing CD40L, IL2, and IL21[27,62] to stimulate antibody production from individual B cells in culture. Together, these techniques permitted the high-throughput isolation and screening of B cells for cross-neutralization.

Targeting functionally conserved epitopes between homologous viruses is an attractive strategy to reduce the risk of developing drug resistance due to the emergence of escape mutations; and, at the same time, can increase the benefit of pre-exposure prophylaxis by protecting against a broader array of pathogens. The cross-neutralizing 3 × 1 mAb binds a site on HPIV3 preF, which we called site X, located between the equator and apex, on the vertices of the prefusion F trimer. It is possible that 3 × 1 binds the furin cleavage site of HPIV3, specifically the C-terminus of the F2 protein, although this region was too disordered to model effectively. Antibodies that cross-neutralize phylogenetically related viruses tend to target well-conserved epitopes[63]. The interaction between non-polar protrusions on the heavy chain of 3 × 1 with grooves on the surface could represent one mechanism that allows binding to both HPIV3 and HPIV1. Further, the light chain of 3 × 1 encompasses a large protrusion on HPIV3 preF and forms a hydrogen bond at Asp143, which is conserved in HPIV1. 3 × 1 likely binds the HRA helix of both HPIV3 and HPIV1, which is a site that undergoes significant motion during fusion as the F protein transitions from the preF to the postF conformation. Further structural analysis

will be needed to develop a better understanding of the molecular interactions that mediate 3 × 1 neutralization of HPIV1.

The MxR mAb neutralizes both HMPV and RSV and shares notable similarities with another antibody, MPE8[14,34], including significant sequence similarity that leads to comparable modes of binding. However, MxR is a more somatically hypermutated antibody and facilitates site III recognition without the extended CDRH3 seen in MPE8, which could provide more energetically favorable site recognition. This structural difference may be the basis for the observed difference between MxR and MPE8 in their neutralization of HMPV, with MxR having ~5.7-fold greater potency.

Palivizumab is currently the only FDA-approved antibody for the prevention of RSV in high-risk infants. However, because the protection afforded by palivizumab is restricted to RSV, it has not gained widespread use for immunocompromised children or adults in whom other viruses like HMPV, HPIV3, and HPIV1 contribute significantly to disease[64]. A cocktail of MxR and 3 × 1 could therefore potentially fulfill this unmet need for a more broadly protective drug. The clinical indication for the RSV-specific D25 monoclonal antibody, which is being developed as nirsevimab to replace palivizumab, is similarly focused on infants[33]. We and others have observed reduced binding of D25 to the preF protein of RSV subtype B compared to RSV subtype A [Fig. 3c; also ref. [65]]. This is notable because escape mutations to D25 have been identified in infants who suffered a breakthrough infection with RSV subtype B after receiving nirsevimab[32]. RB1 is another antibody in clinical development with equal potency against RSV-A and -B but does not neutralize HMPV[35]. The cross-neutralizing MxR antibody we describe in the present study binds strongly to the preF proteins of both RSV subtypes A and B and also neutralizes HMPV. An analysis of potential escape mutations that could arise clinically and their fitness cost to viral replication is an important next step in the development of MxR and 3 × 1.

To investigate the potential clinical utility of administering a cocktail of MxR and 3 × 1 to protect against RSV, HMPV, HPIV3, and HPIV1, we focused our in vivo efficacy studies on immunoprophylaxis. The importance of preventing respiratory viral infections for vulnerable populations has become increasingly apparent during the COVID-19 pandemic. A cocktail of two SARS-CoV-2- specific monoclonal antibodies marketed as Evusheld was authorized by the FDA for prophylaxis in immunocompromised individuals who are expected to mount a poor response to vaccination. In a phase III trial, Evusheld, administered intramuscularly as 600 mg of total antibody, led to an 83% relative risk reduction in symptomatic COVID-19[66]. Due to escape mutations present in the omicron variant, the FDA revised the dose to 1200 mg of total antibody which, for a 60 kg individual, is a 20 mg/kg dose. In our in vivo efficacy studies, we similarly administered a cocktail of two antibodies MxR and 3 × 1, each at 5 mg/kg for a total 10 mg/kg dose. Therefore, the doses tested in the present study are within the range of other antibodies already in clinical use, leaving room for increased dosing in future human studies. Together, MxR and 3 × 1 represent promising mAb candidates for further development to protect against a broad array of respiratory viral infections in highly vulnerable patient populations.

## Methods

### Study design
This study complies with all relevant ethical regulations and was reviewed and approved by the Fred Hutchinson Cancer Center Institutional Review Board. Peripheral blood was obtained by venipuncture from healthy, HIV-seronegative adult volunteers enrolled in the Seattle Area Control study after informed consent (Protocol #5567). PBMCs were isolated from whole blood using Accuspin System Histopaque-1077 (Sigma-Aldrich, cat#10771). Studies involving human spleens were deemed non-human subjects research since tissue was de-identified. Spleen samples were deemed non-human subjects research

by the Fred Hutch Institutional Review Board and as defined by the Common Rule from the Office for Human Research Protections. Tissue was de-identified and originated from deceased donors in which the spleen would have otherwise been discarded during procurement of other organs (i.e., liver) for donation. Tissue fragments were passed through a basket screen, centrifuged at 300×g for 7 min, incubated with ACK lysis buffer (Thermo Fisher, cat#A1049201) for 3.5 min, resuspended in RPMI (Gibco, cat#11875093), and passed through a stacked 500 and 70 µm cell strainer. Cells were resuspended in 10% dimethylsulfoxide in heat-inactivated fetal calf serum (Gibco, cat#16000044) and cryopreserved in liquid nitrogen before use.

### Cell lines
293F cells (Thermo Fisher, cat#R79007) were cultured in Freestyle 293 media (Thermo Fisher, cat#12338026). Vero cells (ATCC CCL-81), LLC-MK2 cells (ATCC CCL-7.1), and HEp-2 (ATCC CCL-23) were cultured in DMEM (Gibco, cat#12430054) supplemented with 10% fetal bovine serum and 100 U/mL penicillin plus 100 µg/mL streptomycin (Gibco, cat#15140122). Although HEp-2 is a commonly mis-identified cell line due to HeLa cell contamination (iclac.org/databases/cross-contaminations/), HEp-2 is traditionally used to grow RSV to high titers.

### Viruses
The recombinant viruses RSV-GFP, HMPV-GFP, HPIV1-GFP, and HPIV3-GFP have been previously described[46,67–69] and were modified, respectively, from RSV strain A2 (GenBank accession number KT992094), HMPV CAN97-83 (GenBank accession number AY297749), HPIV1/Washington/20993/1964 (GenBank accession number AF457 102), and HPIV3 JS (GenBank accession number Z11575) to express enhanced GFP. RSV subtype B strain 18537 (GenBank accession number MG813995) was obtained from ATCC (cat#VR-1580). RSV subtype B strain B1 (GenBank accession number AF013254.1) was obtained from ViraTree (cat#RSVB-GFP3). HMPV, HPIV1, and HPIV3 were cultured on LLC-MK2 cells, and RSV was cultured on HEp-2 cells. Virus was purified by centrifugation in a discontinuous 30%/60% sucrose gradient with 0.05 M HEPES and 0.1 M MgSO$_4$ (Sigma-Aldrich, cat#H4034 and 230391, respectively) at 120,000×g for 90 min at 4 °C. Virus titers were determined by infecting Vero cell monolayers in 24-well plates with serial 10-fold dilutions of the virus, overlaying with DMEM containing 0.8% methylcellulose (Sigma-Aldrich, cat#M0387). For assays involving HPIV1 and HMPV, 1.2% of 0.05% of Trypsin (Gibco, cat#25300054) was included in the media. Fluorescent plaques were counted using a Typhoon scanner (GE Life Sciences) at 5 days post-infection.

### Expression and purification of antigens
Expression plasmids for His-tagged RSV, HMPV, and HPIV3 preF and postF antigens are previously described[26,70,71]. 293F cells were transfected at a density of $10^6$ cells/mL in Freestyle 293 media using 1 mg/mL PEI Max (Polysciences, cat#24765). Transfected cells were cultured for 7 days with gentle shaking at 37 °C. Supernatant was collected by centrifuging cultures at 2500×g for 30 min followed by filtration through a 0.2 µM filter. The clarified supernatant was incubated with Ni Sepharose beads overnight at 4 °C, followed by washing with wash buffer containing 50 mM Tris, 300 mM NaCl, and 8 mM imidazole. His-tagged protein was eluted with an elution buffer containing 25 mM Tris, 150 mM NaCl, and 500 mM imidazole. The purified protein was run over a 10/300 Superose 6 size exclusion column (GE Life Sciences, cat#17−5172−01). Fractions containing the trimeric F proteins were pooled and concentrated by centrifugation in a 50 kDa Amicon ultra-filtration unit (Millipore, cat#UFC805024). The concentrated sample was stored in 50% glycerol at −20 °C.

### Tetramerization of antigens
Purified F antigens were biotinylated using an EZ-link Sulfo-NHS -LC-Biotinylation kit (Thermo Fisher, cat#A39257) with a 1:1.3 molar ratio of

biotin to F. Unconjugated biotin was removed by centrifugation using a 50 kDa Amicon Ultra size exclusion column (Millipore). To determine the average number of biotin molecules bound to each molecule of F, streptavidin-PE (ProZyme, cat#PJRS25) was titrated into a fixed amount of biotinylated F at increasing concentrations and incubated at room temperature for 30 min. Samples were run on an SDS–PAGE gel (Invitrogen, cat#NW04127BOX), transferred to nitrocellulose, and incubated with streptavidin–Alexa Fluor 680 (Thermo Fisher, cat#S32358) at a dilution of 1:10,000 to determine the point at which there was excess biotin available for the streptavidin–Alexa Fluor 680 reagent to bind. Biotinylated F was mixed with streptavidin–allophycocyanin (APC) or streptavidin–PE at the ratio determined above to fully saturate streptavidin and incubated for 30 min at room temperature. Unconjugated F was removed by centrifugation using a 300 K Nanosep centrifugal device (Pall Corporation, cat#OD300C33). APC/DyLight755 and PE/DyLight650 tetramers were created by mixing F with streptavidin–APC pre–conjugated with DyLight755 (Thermo Fisher, cat#62279) or streptavidin–PE pre-conjugated with DyLight650 (Thermo Fisher, cat#62266), respectively, following the manufacturer's instructions. On average, APC/DyLight755 and PE/DyLight650 contained 4–8 DyLight molecules per APC and PE. The concentration of each tetramer was calculated by measuring the absorbance of APC (650 nm, extinction coefficient = $0.6 \, \mu M^{-1} \, cm^{-1}$) or PE (566 nm, extinction coefficient = $2.0 \, \mu M^{-1} \, cm^{-1}$).

## Tetramer enrichment

$1-2 \times 10^8$ frozen PBMCs or $4-8 \times 10^7$ frozen spleen cells were thawed into DMEM with 10% fetal calf serum and 100 U/mL penicillin plus 100 μg/mL streptomycin. Cells were centrifuged and resuspended in 50 μL of ice-cold fluorescence-activated cell sorting (FACS) buffer composed of phosphate-buffered saline (PBS) and 1% newborn calf serum (Thermo Fisher, cat#26010074). PostF APC/DyLight755 and PE/Dylight650 conjugated tetramers were added at a final concentration of 25 nM in the presence of 2% rat and mouse serum (Thermo Fisher) and incubated at room temperature for 10 min. PreF APC and PE tetramers were then added at a final concentration of 5 nM and incubated on ice for 25 min, followed by a 10 mL wash with ice-cold FACS buffer. Next, 50 μL each of anti-APC-conjugated (Miltenyi Biotec, cat#130-090-855) and anti-PE-conjugated (Miltenyi Biotec, cat#130-048-801) microbeads were added and incubated on ice for 30 min, after which 3 mL of FACS buffer was added and the mixture was passed over a magnetized LS column (Miltenyi Biotec, cat#130-042-401). The column was washed once with 5 mL ice-cold FACS buffer and then removed from the magnetic field and 5 mL ice-cold FACS buffer was pushed through the unmagnetized column twice using a plunger to elute the bound cell fraction.

## Flow cytometry

Cells were incubated in 50 μL of FACS buffer containing a cocktail of antibodies for 30 min on ice prior to washing and analysis on a FACS Aria (BD). Antibodies included anti-IgM FITC (G20–127, BD, cat#555782, 1:80 dilution), anti-CD19 BUV395 (SJ25C1, BD, cat#563551, 1:20 dilution), anti-CD3 BV711 (UCHT1, BD, cat#563725, 1:50 dilution), anti-CD14 BV711 (M0P-9, BD, cat#563372, 1:50 dilution), anti-CD16 BV711 (3G8, BD, cat#563127, 1:50 dilution), anti-CD20 BUV737 (2H7, BD, cat#612849, 1:20 dilution), anti-IgD BV605 (IA6–2, BD, cat#563313, 1:50 dilution), anti-CD27 PE/Cy7 (LG.7F9, eBioscience, cat#25-0271-82, 1:160 dilution), and a fixable viability dye (Tonbo Biosciences, cat#13-0870-T500, 1:250 dilution). B cells were individually sorted into either 1) empty 96-well PCR plates and immediately frozen, or 2) flat bottom 96-well plates containing feeder cells that had been seeded at a density of 28,600 cells/well one day prior in 100 μL of IMDM media (Gibco, cat#31980030) containing 10% fetal calf serum, 100 U/mL penicillin plus 100 μg/mL streptomycin, and 2.5 μg/mL amphotericin. B cells sorted onto feeder cells were cultured at 37 °C for 13 days.

## B cell receptor sequencing

For individual B cells sorted and frozen into empty 96-well PCR plates, reverse transcription (RT) was directly performed after thawing plates using SuperScript IV (Thermo Fisher, cat#18090200)[72,73]. Briefly, 3 μL RT reaction mix consisting of 3 μL of 50 μM random hexamers (Thermo Fisher, cat#48190011), 0.8 μL of 25 mM deoxyribonucleotide triphosphates (dNTPs; Thermo Fisher, cat#N8080261), 1 μL (20 U) SuperScript IV RT, 0.5 μL (20 U) RNaseOUT (Thermo Fisher, cat#10777019), 0.6 μL of 10% Igepal (Sigma-Aldrich, cat#I8896), and 15 μL RNase-free water was added to each well containing a single sorted B cell and incubated at 50 °C for 1 h. For individual B cells sorted onto feeder cells, the supernatant was removed after 13 days of culture, plates were immediately frozen on dry ice, stored at −80 °C, thawed, and RNA was extracted using the RNeasy Micro Kit (Qiagen, cat#74034). The entire eluate from the RNA extraction was used instead of water in the RT reaction. Following RT, 2 μL of cDNA was added to 19 μL PCR reaction mix so that the final reaction contained 0.2 μL (0.5 U) HotStarTaq Polymerase (Qiagen, cat#203607), 0.075 μL of 50 μM 3′ reverse primers, 0.115 μL of 50 μM 5′ forward primers, 0.24 μL of 25 mM dNTPs, 1.9 μL of 10 × buffer (Qiagen), and 16.5 μL of water. The PCR program was 50 cycles of 94 °C for 30 s, 57 °C for 30 s, and 72 °C for 55 s, followed by 72 °C for 10 min for heavy and kappa light chains. The PCR program was 50 cycles of 94 °C for 30 s, 60 °C for 30 s, and 72 °C for 55 s, followed by 72 °C for 10 min for lambda light chains. After the first round of PCR, 2 μL of the PCR product was added to 19 μL of the second-round PCR reaction so that the final reaction contained 0.2 μL (0.5 U) HotStarTaq Polymerase, 0.075 μL of 50 μM 3′ reverse primers, 0.075 μL of 50 μM 5′ forward primers, 0.24 μL of 25 mM dNTPs, 1.9 μL 10 × buffer, and 16.5 μL of water. PCR programs were the same as the first round of PCR. 4 μL of the PCR product was run on an agarose gel to confirm the presence of a ~500-bp heavy chain band or ~450-bp light chain band. 5 μL from the PCR reactions showing the presence of heavy or light chain amplicons was mixed with 2 μL of ExoSAP-IT (Thermo Fisher, cat#78201) and incubated at 37 °C for 15 min followed by 80 °C for 15 min to hydrolyze excess primers and nucleotides. Hydrolyzed second-round PCR products were sequenced by Genewiz with the respective reverse primer used in the second-round PCR, and sequences were analyzed using IMGT/V-Quest to identify V, D, and J gene segments. Paired heavy chain VDJ and light chain VJ sequences were cloned into pTT3-derived expression vectors containing the human IgG1, IgK, or IgL constant regions using In-Fusion cloning (Clontech, cat#638911)[74].

## Monoclonal antibody production

Secretory IgG was produced by co-transfecting 293F cells at a density of $10^6$ cells/mL with the paired heavy and light chain expression plasmids at a ratio of 1:1 in Freestyle 293 media using 1 mg/mL PEI Max. Transfected cells were cultured for 7 days with gentle shaking at 37 °C. Supernatant was collected by centrifuging cultures at 2500×$g$ for 15 min followed by filtration through a 0.2 μM filter. Clarified supernatants were then incubated with Protein A agarose (Thermo Scientific, cat#22812) followed by washing with IgG-binding buffer (Thermo Scientific, cat#21007). Antibodies were eluted with IgG Elution Buffer (Thermo Scientific, cat#21004) into a neutralization buffer containing 1 M Tris-base pH 9.0. Purified antibody was concentrated and the buffer was exchanged into PBS using an Amicon ultrafiltration unit with a 50 kDa molecular weight cutoff.

## Fab preparation

Fab was produced by incubating 10 mg of IgG with 10 μg of LysC (New England Biolabs, cat#P8109S) overnight at 37 °C followed by incubating with protein A for 1 h at room temperature. The mixture was then centrifuged through a PVDF filter, concentrated in PBS with a 30 kDa Amicon Ultra size exclusion column, and purified further by size-exclusion chromatography (SEC) using Superdex 200 (GE

Healthcare Life Sciences, cat#17–5175–01) in 5 mM HEPES and 150 mM NaCl.

## Bio-layer interferometry

Bio-layer interferometry (BLI) assays were performed on the Octet.Red instrument (ForteBio) at room temperature with shaking at 500 rpm. For apparent affinity ($K_D$) analyses, anti-penta His capture sensors (ForteBio, cat#18–5120) were loaded in kinetics buffer (PBS with 0.01% bovine serum albumin, 0.02% Tween 20, and 0.005% $NaN_3$, pH 7.4) containing 0.5 μM purified RSV-A, RSV-B, HMPV, or HPIV3 preF for 150 s. After loading, the baseline signal was recorded for 60 s in a kinetics buffer. The sensors were then immersed in kinetics buffer containing 266.7, 133.3, 66.7, 33.3, or 16.7 nM of purified monoclonal antibody for a 300 s association step followed by immersion in kinetics buffer for a dissociation phase of at least 600 s. The maximum response was determined by averaging the nanometer shift over the last 5 s of the association step after subtracting the background signal from each analyte-containing well using a negative control mAb at each time point. Curve fitting was performed using a 1:1 binding model and ForteBio Octet data analysis software release 9.0. For competitive binding assays, anti-penta His capture sensors were loaded in kinetics buffer containing 1 μM His-tagged HPIV3 preF for 300 s. After loading, the baseline signal was recorded for 30 s in kinetics buffer. The sensors were then immersed for 300 s in kinetics buffer containing 40 μg/mL of the first antibody followed by immersion for another 300 s in kinetics buffer containing 40 μg/mL of the second antibody. Percent competition was determined by dividing the maximum increase in signal of the second antibody in the presence of the first antibody by the maximum signal of the second antibody alone.

## Fusion inhibition assay

Vero cells were plated into 96-well flat bottom plates in duplicate and cultured for 48 h then incubated with HPIV3 at an MOI = 0.01 for one hour at 37 °C. Cells were washed five times to remove the un-adsorbed virus. 3 × 1 (10 μg/mL) or media (control) was added to cells. Cell-to-cell spread and syncytia formation were examined at days 1, 3, and 5 post-infection using an EVOS Cell Imaging System (Thermo Fisher).

## Neutralization assays

For neutralization screening of culture supernatants, Vero cells were seeded in 96-well flat bottom plates and cultured for 48 h. After 13 days of culture, 40 μL of B cell culture supernatant was mixed with 25 μL of sucrose-purified GFP-HPIV1 diluted to 2000 plaque-forming units (pfu)/mL for 1 h at 37 °C. Vero cells were then incubated with 50 μL of the supernatant/virus mixture for 1 h at 37 °C to allow viral adsorption. Next, each well was overlaid with 100 μL DMEM containing 0.8% methylcellulose and 1.2% of 0.05% trypsin. Fluorescent plaques were counted at 5 days post-infection using a Typhoon imager.

Neutralizing titers of monoclonal antibodies were determined by a 60% plaque reduction neutralization test ($PRNT_{60}$). Vero cells were seeded in 24-well plates and cultured for 48 h. Monoclonal antibodies were serially diluted 1:4 in 120 μL DMEM and mixed with 120 μL of sucrose-purified RSV, HMPV, HPIV3, or HPIV1 diluted to 2000 pfu/mL for one hour at 37 °C. Vero cells were incubated with 100 μL of the antibody/virus mixture for 1 h at 37 °C to allow viral adsorption. Each well was then overlaid with 500 μL DMEM containing 0.8% methylcellulose. Fluorescent plaques were counted at 5 days post-infection using a Typhoon imager. $PRNT_{60}$ titers were calculated by linear regression analysis (http://exon.niaid.nih.gov/plaquereduction/).

## Cryo-EM complex and grid preparation

1.47 mg of RSV preF with 1.45 mg of MxR Fab were mixed and incubated overnight at 4 °C with gentle rocking, before SEC purification over a 10/300 Superose 6 column (Cytiva, cat#29091596). A very broad peak eluted, with the nine largest fractions concentrated to 0.22 mg/mL using a 10 kDa cutoff Amicon ultrafiltration unit (Sigma-Aldrich, cat#UFC8030). For 3 × 1, we incubated 50 μg of HPIV3 preF with 150 μg of 3 × 1 Fab overnight at 4 °C with gentle rocking before SEC purification over a Superdex 200 16/600 SEC column (Cytiva, cat#28989335). A single narrow high molecular weight peak was eluted and was concentrated to 0.4 mg/mL using a 10 kDa cutoff Amicon ultrafiltration unit.

Both complexes were frozen on Quantifoil 1.2/1.3 UltrAu 300 mesh grids (Electron Microscopy Sciences, cat#Q350AR13A) using a Vitrobot Mk. IV (Thermo Fisher) at 22 °C and 100% humidity. Dodecyl-β-D-maltoside (DDM) was added to the sample (0.05% final concentration) 30 min prior to the start of freezing, and a PELCO easiGlow™ (Tedpella, Cat#91000) was used to glow discharge the grids. The final MxR:RSV grids used for the collection were frozen with 4 μL of the sample at 0.21 mg/mL, a 14 s blot time, 0 blot force, and a 5 s wait between application and blotting. The final 3 × 1:HPIV3 grids used for the collection were frozen with 4 μL of the sample at 0.19 mg/mL, a 12 s blot time, 0 blot force, and a 15 s wait between application and blotting.

## Cryo-EM data collection, processing, and model refinement

Datasets were collected on a Titan Krios G3 cryo-electron microscope (Thermo Fisher Scientific) operating at 300 kV with a K3 DED (Gatan Inc.) at the Pacific Northwest Center for Cryo-EM (PNCC). Data were collected in 50 frame movies using Serial EM at ×92,000 magnification (0.514425 Å/px using super-resolution mode). Both collections ran for 24 h, producing 6282 movies for 3 × 1:HPIV3 and 5796 movies for MxR:RSV. 3 × 1:HPIV3 was collected at a 30° tilt due to the observed preferred orientation during screening.

Data sets were processed using cryoSPARC v3.3.1[75]. Following the import, patch motion correction (micrographs binned to 1.02885 Å/px), and contrast transfer function (CTF) estimate, micrographs were curated for <4 Å CTF fit. Blob picker was used to select ~50,000 particles, which underwent 2D classification to produce templates for template-based picking. These picks were inspected, curated, and extracted (192-pixel box size at 2.0577 Å/px) with 1.49 million particles for MxR:RSV and 2.02 million particles for 3 × 1:HPIV3.

Following two rounds of 2D classification, 100 classes each, we were left with 377,982 MxR:RSV particles showing three Fabs bound. A single ab-initio model was generated and refined. Particles were re-extracted at 1.02885 Å/px and subject to local CTF refinement and non-uniform refinement with C3 symmetry. Following this, particles were further curated and re-extracted using Local Motion Correction, producing 354,958 particles. Non-uniform refinement with a custom mask cropping out the CH1 region and GCN4 domain produced a sharpened map (GSFSC = 0.143) of 2.41 Å resolution using C1 symmetry, and 2.24 Å using C3 symmetry.

The template generation for 3 × 1:HPIV3 produced a mix of classes with either one, two, or three Fabs bound, all of which were used for original template picking. We performed one round of 2D classification with 100 classes, selecting solitary particles which showed any number of Fabs bound, producing a stack of 1.3 million particles. Ab-initio modeling produced a single map with one Fab bound to HPIV3 preF, and subsequent refinement produced a map with a GSFSC resolution of 3.7 Å at the 0.143 cutoffs using C1 symmetry. An additional three Fab map was produced with C3 symmetry in non-uniform refinement to 4.3 Å, though due to the low resolution, this map was not used in model building. Minor improvements were made by performing a C3 homogenous refinement, followed by C3 symmetry expansion, 3D classification with four classes, and the removal of duplicate particles from the most homologous class, producing a stack of 1.04 million particles. Particles were re-extracted with the Local Motion Correction job, binned to 1.02885 Å/px, and the map refined using C1 non-uniform refinement. This produced our final sharpened map with a resolution of 3.62 Å (GSFSC = 0.143).

Both final maps were further processed using the COSMIC[2] computer[76] with the DeepEMhancer[77] module. Both the DeepEM enhanced and cryoSPARC sharp maps were used in fitting the structures of MxR:RSV and 3×1:HPIV3 to the map. Structure refinement and validation were done in the Phenix[78] software suite and Coot[79]. Further refinement was done in ChimeraX[80] using the ISOLDE[81] plugin as necessary. The unmasked half maps, sharpened maps, and the masks used were deposited to the PDB and EMDB. All graphics were produced in Pymol[82]. Graphs were prepared using GraphPad Prism 9. Buried surface area analysis was carried out using the PDBePISA[83] server.

## Animals and viral challenge

All procedures were reviewed and approved by the Fred Hutch Institutional Animal Care and Use Committee and conducted in accordance with institutional and National Institutes of Health guidelines. Golden Syrian hamsters (*Mesocricetus auratus*) were infected intranasally with 100 μL of $10^5$ pfu RSV, HMPV, HPIV3, or HPIV1. Sample sizes (overall $n = 24$ for serum antibody concentration experiments, $n = 96$ for single infection experiments, and $n = 24$ for co-infection experiments) were consistent with previously published experiments testing the efficacy of RSV monoclonal antibodies in the cotton rat model[58,59,84,85]. Only male animals (4–8 weeks of age) were assessed in this work, because no association between sex and clinical outcomes has been observed in adults with RSV, HMPV, HPIV3, or HPIV1[86,87]. Monoclonal antibody was administered intramuscularly at 5, 2.5, or 1.25 mg/kg in 50 μL PBS 2 days prior to infection. For the co-infection model, $10^5$ pfu of each virus was mixed in 100 μL 1× DPBS, and 5 mg/kg of each monoclonal antibody was mixed in 50 μL 1× DPBS. Nasal turbinates and lungs were removed for viral titration by plaque assay 5 days post-infection and clarified by centrifugation in DMEM. Confluent Vero cell monolayers were inoculated in duplicate with diluted homogenates in 24-well plates. After incubating for 1 h at 37 °C, wells were overlaid with 0.8% methylcellulose (made with 1.2% of 0.05% trypsin for specimens from animals challenged with HMPV and HPIV1). After 5 days, plaques were counted using the Typhoon imager to determine titers as pfu per gram of tissue. Aliquots of nasal turbinate and lung samples were also saved for the quantification of viral load by real-time PCR.

## ELISA

Serum concentrations of MxR and 3×1 were measured by ELISA. Briefly, Nunc maxsorp 96-well plates (Thermo Fisher, cat#442404) were coated with 100 ng of goat anti-human Fab (Jackson ImmunoResearch, cat#109-005-097) for 90 min at 4 °C. Wells were washed three times with 1×DPBS and then incubated with 1×DPBS containing 1% bovine serum albumin (Sigma-Aldrich, cat#A2153) for 1 h at room temperature. Antigen-coated plates were incubated with serum for 90 min at 4 °C. A standard curve was generated with serial two-fold dilutions of palivizumab. Wells were washed three times with 1×DPBS followed by a one-hour incubation with horseradish peroxidase-conjugated goat anti-human total Ig at a dilution of 1:6000 (Invitrogen, cat#31412). Wells were then washed four times with 1×DPBS followed by a 5–15 min incubation with TMB substrate (SeraCare, cat#5120-0053). Absorbance was measured at 405 nm using a Softmax Pro plate reader (Molecular Devices). The concentration of antibody in each sample was determined by reference to the standard curve and dilution factor.

## Real-time PCR

Viral RNA was extracted from 140 μL of sample homogenate using the QIAamp vRNA mini kit (Qiagen, cat#52904). RNA was eluted with 50 μL water and 11 μL of the eluate was used for reverse transcription. Custom reverse transcription primers for RSV (5′-TCCAGCAAATAC ACCATCCAAC-3′) and HPIV3 (5′-CTAGAAGGTCAAGAAAAGGGAACT

C-3′) were designed to specifically bind to the genomes of RSV and HPIV3, respectively. One microliter of each primer at 2 μM was included in a RT reaction mix containing 1 μL of RNaseOut, 1 μL of 0.1 M DTT, 4 μL of SuperScript IV buffer, and 1 μL of SuperScript IV reverse transcriptase (Thermo Fisher, cat#18090200). Reverse transcription was performed with the following cycle: 42 °C for 10 min, 20 °C for 10 min, 50 °C for 10 min, and 80 °C for 10 min. Custom TaqMan Gene Expression Assays were developed for RSV (forward primer 5′-TGACTCTCCTGATTGTGGGATGATA-3′, reverse primer 5′-CGGCTG TAAGACCAGATCTGT-3′, and reporter 5′-CCCCTGCTGCTAATTT-3′) & HPIV3 (forward primer, 5′-CGGTGACACAGTGGATCAGATT-3′, reverse primer 5′-TGTTTCAACCATAAGAGTTACCAAGCT-3′, and reporter 5′-ACCGCATGATTGACCC-3′). The PCR reaction consisted of 2.5 μL of these primers, 10 μL of the reverse transcription reaction, 25 μL of TaqMan Universal Master Mix II with UNG (Thermo Fisher, cat#4440038), and 12.5 μL water. Real-time PCR was performed using the QuantStudio 7 Flex Real-Time PCR System with the following parameters: 50 °C for 2 min and 95 °C for 10 min followed by 40 cycles at 95 °C for 15 s and 60 °C for 1 min. To generate a standard curve, viral RNA was extracted from sucrose-purified viral stocks of RSV with known titers in pfu/mL. Reverse transcription was performed as above. The reverse transcription reaction was serially diluted eight times at 1:4 in water. Real-time PCR was performed as above, and standard curves were generated to interpolate viral loads to pfu/g using QuantStudio Real-time PCR Software v1.0.

## Statistical analysis

Statistical analysis was performed using GraphPad Prism 7. Pairwise statistical comparisons were performed using Mann–Whitney two-tailed testing. $p < 0.05$ was considered statistically significant. Data points from individual samples are displayed.

## Reporting summary

Further information on research design is available in the Nature Portfolio Reporting Summary linked to this article.

## Data availability

Sequencing and structural data that support the findings of this study have been deposited in the Protein Data Bank (PDB) and Electron Microscopy Data Bank (EMDB) and are accessible through accession numbers PDB 8DG8 (EMDB 27418) for 3×1/HPIV3 and PDB 8DG9 (EMDB 27419) for MxR/RSV. Source data are provided with this paper.

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

## Acknowledgements

We thank Julie McElrath for PBMCs from the Seattle Area Control cohort; Leo Stamatatos for the use of laboratory space and equipment; Andrew McGuire for providing CD40L/IL2/IL21-expressing 3T3 cells; Ramasamy Bakthavatsalam and LifeCenter Northwest for providing de-identified spleen remnants; Ursula Buchholz, Shirin Munir, and Peter Collins for providing the GFP-expressing RSV, HMPV, HPIV3, and HPIV1; Peter Kwong, Guillaume Stewart-Jones, and Aliaksandr Druz for expression of HPIV3 preF; Barney Graham for expression of RSV preF; Theodore Jardetzky and Xiaolin Wen for expression of HMPV preF; Steve Voght for

proof-reading the manuscript; Paula Culver, Francesca Urselli, and Laura Yates for administrative support; the Fred Hutchinson Flow Cytometry Shared Resources for assistance with instruments; the Fred Hutchinson Comparative Medicine Shared Resources for assistance with housing hamsters; and the members of the Taylor Lab and Boonyaratanakornkit Lab for helpful discussions. Experimental schematics were created with BioRender.com. This study was supported by the Vaccine and Infectious Disease Division Faculty Initiative (J.B. and J.J.T.) and Evergreen Beyond Pilot Award (J.B. and J.J.T.) from the Fred Hutchinson Cancer Center, a sponsored research agreement with IgM Biosciences (J.B. and J.J.T.), a New Investigator Award from the American Society for Transplantation and Cellular Therapy (J.B.), and the Amy Strelzer Manasevit Award from the National Marrow Donor Program (J.B.). A portion of this research was supported by U24 GM129547 from the NIH and performed at the PNCC at OHSU and accessed through EMSL (grid.436923.9), a DOE Office of Science User Facility sponsored by the Office of Biological and Environmental Research. Further electron microscopy data was generated using the Fred Hutchinson Cancer Center Electron Microscopy Shared Resource, which is supported in part by P30 CA015704. The content is solely the responsibility of the authors and does not necessarily represent the official views of the National Institutes of Health.

## Author contributions

M.C. and M.B. designed and conducted the experiments, analyzed the data, and wrote the manuscript. M.D.G. conducted experiments, analyzed data, and edited the manuscript. J.V.R. and M.P. coordinated and performed the structural analysis and wrote the manuscript. J.J.T. conceived the study, designed experiments, analyzed the data, and edited the manuscript. J.B. conceived the study, designed and conducted the experiments, analyzed the data, and wrote the manuscript.

## Competing interests

The authors J.B. and J.J.T. are inventors of patent applications filed by Fred Hutchinson Cancer Center directed to the 3 × 1 and MxR antibodies. The remaining authors declare no competing interests.
