## [Peer Review File · Nature Communications]

Cross-Protective Antibodies Against Common Endemic Respiratory VirusesReviewers' Comments:

Reviewer #1:

Remarks to the Author:

This manuscript describes the isolation and characterization of human antibodies targeting the prefusion F proteins of a few related respiratory viruses including RSV, hMPV, PIV1 and PIV3.

The approach is to use single B cell sorting to isolate naturally occurring human monoclonal antibodies from human donors who have been previously infected naturally. The "bait and switch" strategy of binding to one antigen and neutralizing another related virus appears robust in this study for discovery of cross-neutralizing Abs. They performed standard binding and neutralization assays to characterize two cross-reactive antibodies against RSV/hMPV, or PIV1/3, respectively. A lot of work has been done in this field on RSV, and recently a few publications also described profiling of antibody response against human metapneumovirus. Multiple hMPV and RSV cross reactive antibodies also have been described, including those site III antibodies similarly as the MxR Ab described here. Therefore, the most novel data this paper is on the cross-reactive PIV1/3 antibody 3x1.

CryoEM structures were determined to map the binding epitopes and help to understand the antigen/antibody interaction in detail. The EM data is solid. MxR/RSV structure was well analyzed with higher resolution, providing a solid comparison to other RSV/hMPV cross-neutralizing Abs in literature. For the 3x1/PIV3 structure, though the overall resolution is limited, they achieved higher local resolution in the binding site and can therefore map the CDR:antigen interactions to explain the increased breadth of 3x1. However, even with a cryoEM structure of 3x1/PIV3, it is not clear what's the mechanism of neutralization. Authors claimed that it might inhibit prefusion to postfusion conformational change, or bind the furin cleavage site, but provided no experimental evidence. Authors should consider other in vitro functional assays (such as fusion inhibition assay, furin cleavage assay) to further supports the hypothesis. Further, authors should consider generating escape mutants with these antibodies, as it would be important to understand this if these antibodies are considered as candidates for further development.

The authors also performed in vivo experiments to demonstrate the protection efficacy of these antibodies. While the data did suggest that these antibodies could offer protection, the choice of hamster model might not be ideal for certain pathogens. In addition, the lack of dose titration and lack of benchmark antibodies in the in vivo studies prevents comparison of these antibodies to others previously described in the field.

One weakness of this paper is that only a few antibodies are described. Of note, only two B cell clones were selected out of 900. It would be more interesting if the authors could profile more PIV antibodies as knowledge in the field is scarce. In the field of RSV, it is known the prefusion specific Abs tend to be more potent; however, this is not true for hMPV as the equivalent antigenic site 0 is shielded with glycan thus not immune dominant. Little is known for PIV F antibodies, therefore, a comprehensive profiling of PIV antibodies targeting multiple epitopes could add a lot more value.

Specific comments:

P1, line 30: this is not accurate, at least Moderna has announced efforts to develop HMPV/PIV vaccine.

P3, line 9: while prefusion specific antibody might be more potent neutralizer (such as RSV site 0 Abs), there are plenty of examples showing antibodies targeting both preF/postF could also be very potent. Authors should evaluate those B cell clones and might obtain more cross-reactive antibodies that target different epitopes than 3x1.

P3, line 32: why would only one Fab bound to the F trimer for a large portion of the particles, given that one Fab exclusively contacts only one F protomer? I assume there is excess of Fab vs. F? Author might want to discuss if binding one Fab to the trimeric F could potentially change its conformation

and prevent further binding of additional Fabs.

Figure 3d: not clear why D25 is not included in the neutralization assay.

P7, line 1-11: authors discussed potential structural explanations why MxR could bind/neutralize hMPV better than MPE8 and ADI-19425. However, the structures were compared from complexes with RSV F but not hMPV F, while binding to RSV F are similar among these antibodies. CryoEM structures with hMPV F would be more appropriate to offer these insights.

Figure 5: please explain the choice of dose 5 mg/kg. It would be more informative if authors could perform a titration of dose levels and calculate EC90 to establish the required levels of antibody for protection. Also, it is not known what are the concentrations of the antibodies in the serum at day 0, this would allow better comparison among individual animals as well as to literature.

P7, line 41: in the PBS control, PIV3 established much higher titers in the lung compared to PIV1, authors should discuss whether hamster is a good model for PIV1 (and similarly for hMPV). At least for hMPV, higher viral titers have been established in cotton rats (Wyde et al., doi:10.1016/j.antiviral.2004.12.009).

Figure 6: authors described a co-infection model. Because of the plaque assay used can't differentiate RSV vs. PIV3 viruses, it is unclear whether there is interference between the two viruses, i.e. whether both viruses are replicating well when given together. Authors did show qPCR data for the co-infection model, it would be beneficial to compare this data with copy numbers from animals infected with single virus. Ultimately, it would be better to establish plaque assays that differentiate among viruses.

P8, line 53: model PIV1 F structure with 3x1 would be beneficial to understand the conserved mechanism of cross-neutralization. since PIV3 also shares structural homology to RSV/hMPV, authors could discuss if the 3x1 epitope is unique to PIV1/3 or has been previously described for RSV/hMPV antibodies.

P9, line 5: again, the structures of MxR and MPE8 were compared from complexes with RSV F but not hMPV F, while binding to RSV F are similar among these antibodies. CryoEM structures with hMPV F would be more appropriate to offer these insights.

P9, line 15: authors should note that other monoclonal antibodies are also in clinical development and has equal potency against RSV A and B (e.g. Tang et al., <https://doi.org/10.1038/s41467-019-12137-1>).

Reviewer #2:

Remarks to the Author:

The stated goal of these studies was to develop a reagent that would protect against infections of four respiratory pathogens, HPIV1, HPIV3, RSV, and HMPV for, ostensibly, use in immunocompromised populations. The approach was to develop a mAb that cross reacted with the closely related HPIV1 and HPIV3 and a mAb that cross reacted with the closely related RSV and HMPV. The goal was to determine if a cocktail of these two mAb would protect against the four pathogens. While cross reactive mAbs to RSV and HMPV have been reported, none have been developed for the HPIV1 and HPIV3, to my knowledge. The goal of combining such mAbs for protection from four pathogens is moderately novel.

These studies are quite comprehensive and detailed, proceeding from isolation and characterization of the mAb to assays of their protection of animals from challenge.

The authors describe in considerable detail their approaches for finding such antibodies. One protocol was used for developing the human parainfluenza antibody, named 3x1. This task was made more difficult by the failure to isolate a pre-fusion HPIV1 F protein and the authors have devised a clever way around this difficulty. They termed this a "bait-and-switch" strategy. Other investigators might find this approach useful for developing other cross-reacting antibodies. The approach for developing the cross reacting mAb for RSV and HMPV, named MxR, was somewhat more straight forward. The authors cloned and expressed these two antibodies from plasmids and used the purified antibodies to characterize the binding and affinities of the antibodies to three of the target F proteins as well as their neutralizing activities, separately, against each of the four viruses. The authors also characterized the cryo-EM structures of the Fab of mAb 3x1 bound to HPIV3 F protein and cryo-EM structures of the Fab of mAb MxR bound to RSV F protein. Where possible, the authors have compared their results with those of previously isolated mAb to some of the pathogens. While analyses of cryo-EM structures are not complete, missing structures of mAb complexes with HPIV 1 and HMPV F proteins, they do offer a preliminary view of possible mechanisms of neutralization. The authors then proceeded to show that the mAb protected hamsters separately challenged with RSV, HMPV, HPIV1, and HPIV3. Protection from replication of each virus in the lungs was generally good but protection from replication nasal turbinates was negligible. The authors did proceed to test the protection from co-infection with RSV and HPIV3 by a cocktail of MxR and 3x1 showing that upon virus challenge total plaques in lungs (there was no identification of virus in plaques which is quite possible) were considerably reduced, but titers in nasal washes were not reduced significantly. Assay of specific virus load by virus specific qPCR showed less protection. These animal studies are somewhat incomplete and can only be viewed as a preliminary indication of the potential efficacy of the antibody cocktail.

Overall, while these studies are comprehensive, there are some issues that the authors should consider.

1. Some of the figures could be of better quality. Some of the details in the cryo-EM structures described in the text are difficult to impossible to see in the figures. Notably figure 2, panels c-e, and figure 4, panels b-d, should be improved. Colors and printing are hard to see.
2. In the animal experiments, there was no indication of the sex of the hamsters and no consideration of differences in responses in male or female animals. This issue is receiving increased emphasis in study sections as well as journal reviews since there are often differences in responses to immunization and infection in males and females.
3. The animal studies are difficult to interpret since the replication of two different viruses in co-infections has not been characterized. The authors may want to comment on the limitations of these studies here.
4. Did the authors compare protection from challenge in co-infections and single infections with a cocktail of four different mAb each targeting one of the four pathogens with protection using their 3x1 and MxR cocktail? The authors should comment on why their cocktail is better than a 4x cocktail of separate mAb to all four pathogens or a 2x cocktail of an mAb to RSV and a mAb to HPIV3.
5. The authors should comment on the potential of mAb resistant mutants and protection from two viruses.
6. The authors should comment on why cryo-EMs of mAb with HPIV1 and HMPV were not done.

Reviewer #3:
Remarks to the Author:

In this manuscript the authors present the discovery and characterization of two cross-reactive antibodies, which they named 3x1 and MxR. The 3x1 mAb has high neutralization potency against both human parainfluenza virus-1 (HPIV1) and HPIV3 and demonstrates tight binding to the prefusion conformation of the HPIV3 F protein (preF). Similarly, the MxR antibody is highly potent against both respiratory syncytial virus (RSV) and human metapneumovirus (HMPV) and binds the preF proteins of RSV and HMPV with nanomolar or higher affinity. The authors further used cryo-electron microscopy (cryo-EM) to solve structures of Fab 3x1 complexed with preF of HPIV3 and of Fab MxR complexed with PreF of RSV, providing structural basis for tight binding and cross-neutralization. In vivo experiments, the two mAbs were highly efficient in protecting golden Syrian hamsters against the corresponding infections. Importantly, a cocktail of 3x1 and MxR provided protection against HPIV1, HPIV3, RSV and HMPV simultaneously. This makes the combination of 3x1 and MxR a promising candidate for further development, with potential clinical utility in protecting immunocompromised individuals from the four respiratory viruses, which cumulatively account for the majority of lower respiratory tract infections in vulnerable populations. These findings are novel and highly important. I believe this manuscript deserves publication after the issues outlined below are resolved. My comments and questions are mostly focused on validation and presentation of the cryo-EM structures.

Major issues and questions related to structural analysis

1. Some essential cryo-EM validation information is absent from the manuscript. Please include the following in the Supplementary Figures:
 - a. The FSC plot for each map as generated by cryoSPARC, with all the curves present. Currently only the FSC plot of the 3-Fab 3x1 map is included (Fig. S2a), but it appears to be cropped on the right side (unless the data has been binned). Presenting only local resolution "heat maps" is not sufficient.
 - b. Cryo-EM density overlaid with the atomic model for the CDRs that are in contact with preF (for both high-resolution maps).
 - c. Estimates of map-model correlations in some form. Map-model FSC curves (for example, generated with phenix.mtriage) would be helpful.
 - d. A representative micrograph for each complex.
2. The 3x1 mAb has picomolar or even higher affinity to HPIV3 PreF protein (page 3, line 21). Nonetheless (and unlike in the case of the complex between Fab MxR and RSV PreF), size exclusion chromatography apparently isolated mostly non-stoichiometric complexes, with the complex between one Fab and one PreF trimer being the main species according to the cryo-EM results. I have two suggestions:
 - a. To determine whether binding of one 3x1 Fab hinders subsequent Fab binding by altering the global conformation of the PreF protein, the authors should superpose and compare the protomers of HPIV3 PreF in the structure of the complex. Are there any significant differences between the protomers' structures? Furthermore, does the PreF trimer in this complex deviate noticeably from the expected C3 symmetry?
 - b. It would help if the authors measured the affinity of 3x1 Fab and compared it to the affinity of mAb 3x1. It is possible that both Fabs of the antibody need to interact with preF simultaneously for tight binding. If so, the mechanism of neutralization could be more complicated than simply via blocking the furin cleavage site as suggested in the Discussion.
3. The route that the authors took to arrive at the final cryo-EM map of the 3x1 complex with one Fab bound is unusual. Since, according to the Methods and Fig. S1a, the ab initio model generation produced one clear "winner" – with one Fab bound to one trimer – it made sense to refine the population of particles showing one Fab against this model. Why did the authors then choose to impose C3 symmetry on clearly asymmetrical particles and then relax this symmetry, followed by the removal of the particle duplicates that were produced in the process? It seems that doing this should be equivalent to doing nothing and should not change the map. Although the authors state in the Methods (page 12, line 14) that this approach produced minor improvements in the map, the resolution stayed the same (3.6 Å). What was the nature of these improvements and are they quantifiable? I would like to note that refinements of cryo-EM maps in complicated cases often turn into a "kitchen sink," and the path taken by the authors is not by itself disqualifying.
4. On page 4, lines 2-4 of the Results the authors describe hydrogen bonds within Fab 3x1 and

between its residue Asp91 of the light chain and Asp143 of HPIV3 PreF. Is the local resolution sufficient to support this? Can the authors provide close-up view(s) with cryo-EM density or modify their interpretation?

Minor issues and questions related to structural analysis

1. According to Supplementary Table 1, the defocus range for the 3x1 dataset was -0.5 to -3.5 microns, but in the provided validation report it is -1 to -3 microns.
2. According to the Methods (page 11, line 59), the electron exposure was 50 e-/Å², but it is different in the Supplementary Table 1 and the validation reports (~41 e-/Å²).
3. In the Methods the authors use the phrase "with a GSFSC resolution significantly greater than Nyquist at 2.0577 Å/px" (page 12, line 3). Perhaps they meant that the Fourier shell correlation value did not drop to 0 at the Nyquist frequency (which for the binned data should be ~4.4 Å - twice the pixel size; also it is not measured in Å/px)? Strictly speaking, exceeding the Nyquist limit violates a fundamental principle of cryo-EM and beyond.
4. I suggest that the authors carefully revise and polish the cryo-EM Methods. Some issues include:
 - a. Omission of essential references (cryoSPARC, DeepEMhancer, ChimeraX, ISOLDE, Pymol, PDBePISA, ...).
 - b. What is "COSMIC computer" (page 12, line 18)? The reference is to a review about RSV vaccine development.
 - c. Colloquial language not appropriate for a manuscript ("Krios 300kV Cryo-EM scope with a K3 camera") - A Thermo Scientific Titan Krios electron microscope (rather than "scope") operated at 300 kV; K3 is manufactured by Gatan, etc.
 - d. Awkward phrases and sentences:
 - i. "Grids were collected on..." - Datasets were collected on...?
 - ii. "with a super resolution of 0.514425 Å/px" - in the super-resolution mode with a pixel size of...?
 - iii. "The template generation for 3x1:HPIV3 produced a mix of 1 Fab, 2 Fabs, and 3 Fabs..." - produced classes with one, two, and three Fabs bound?
 - iv. "Grids were also glow discharged prior to the start of freezing"
 - v. "A small subset of particles (~50,000) was picked, underwent 2D classification, and classes selected for the template-based picking job."
 - vi. ...and so on
5. Both EMDB validation reports alert about a large discrepancy between author provided resolution and author provided FSC curve (3.62 Å vs 3.98 Å for 3x1; 2.24 Å vs 2.81 Å for RxM). Did the authors deposit the FSC curves obtained without masking? This is not a problem for this manuscript (provided that all the FSC plots will be added as suggested above), but maybe the authors will have a chance to re-upload the FSC plots that support the stated resolutions before finalizing the depositions.
6. The resolution of the RxM complex is impressive (2.2 Å). Did the authors see density for any water molecules? There are no water molecules in the model according to the table.
7. The authors state that they did not build a molecular model for the symmetrical 3x1 map, but it is shown in Fig. S2, panels b-d. Was this the one-Fab model docked into the three-Fab map? This should be stated in the legend.

Other minor issues and questions

1. Fig. 1c: the red bar should be described in the legend.
2. Fig. 2b, bottom panel: PreF protein colors are not defined. The legend only mentions shades of grey.
3. Fig 2e: the rotation angle is 110 degrees, but it is 100 in the legend.
4. Fig. 4a: "dark red" and "light red" are difficult to distinguish; please consider a different combination.
5. Fig. 4a: where is the II' label? Both labels seem to read just II.
6. For consistency with Fig. 2, rotation angles should be indicated above panels 4b-d
7. Figure S4 is not mentioned in the manuscript.
8. A section title states "Identification of a novel cross-neutralizing antibody against multiple parainfluenza viruses" (page 3). Since there are only two such viruses, the title needs to be modified

accordingly.

9. On page 4, line 4 there is a reference to Fig. S2e. There is no panel e in Fig. S2.

We would like to thank the reviewers for their detailed comments and are grateful for the opportunity to respond. The manuscript has been revised and the point-by-point response is found below. Reviewer comments are displayed in regular text, and our responses are in italics. Page and line numbers refer to the manuscript document in which changes are tracked.

Reviewer #1: This manuscript describes the isolation and characterization of human antibodies targeting the prefusion F proteins of a few related respiratory viruses including RSV, hMPV, PIV1 and PIV3.

The approach is to use single B cell sorting to isolate naturally occurring human monoclonal antibodies from human donors who have been previously infected naturally. The “bait and switch” strategy of binding to one antigen and neutralizing another related virus appears robust in this study for discovery of cross-neutralizing Abs. They performed standard binding and neutralization assays to characterize two cross-reactive antibodies against RSV/hMPV, or PIV1/3, respectively. A lot of work has been done in this field on RSV, and recently a few publications also described profiling of antibody response against human metapneumovirus. Multiple hMPV and RSV cross reactive antibodies also have been described, including those site III antibodies similarly as the MxR Ab described here. Therefore, the most novel data this paper is on the cross-reactive PIV1/3 antibody 3x1.

1. CryoEM structures were determined to map the binding epitopes and help to understand the antigen/antibody interaction in detail. The EM data is solid. MxR/RSV structure was well analyzed with higher resolution, providing a solid comparison to other RSV/hMPV cross-neutralizing Abs in literature. For the 3x1/PIV3 structure, though the overall resolution is limited, they achieved higher local resolution in the binding site and can therefore map the CDR:antigen interactions to explain the increased breadth of 3x1. However, even with a cryoEM structure of 3x1/PIV3, it is not clear what's the mechanism of neutralization. Authors claimed that it might inhibit prefusion to postfusion conformational change, or bind the furin cleavage site, but provided no experimental evidence. Authors should consider other in vitro functional assays (such as fusion inhibition assay, furin cleavage assay) to further supports the hypothesis. Further, authors should consider generating escape mutants with these antibodies, as it would be important to understand this if these antibodies are considered as candidates for further development.
 - a. *We are grateful to the reviewer for the careful reading of the manuscript. As recommended by the reviewer, we have added data from a fusion inhibition assay. Using fluorescence microscopy, we show that 3x1 blocks cell-to-cell spread and syncytia formation by HPIV3 in vitro (Fig. S1). We considered performing a furin cleavage assay, but the binding assays and structural data indicate that 3x1 binds to the mature prefusion conformation of the F protein following furin cleavage (Fig. S7). Therefore, 3x1 would not be expected to inhibit furin cleavage. We have added a discussion of these methods and results (Page 3 lines 26-27).*
 - b. *We also agree with the reviewer that escape mutations are important to consider for further clinical development. We have included a discussion of escape mutations (Page 10 lines 53-54), but this analysis is out of the scope of this manuscript.*
 - c. *To provide a more specific mechanism of how 3x1 blocks fusion, we have included additional structural analysis indicating that 3x1 binds to heptad repeat A, which undergoes significant rearrangement in the transition from the prefusion to postfusion conformation. As discussed in our response below to point #12:
 - i. *“The 3x1 mAb targets the heptad repeat epitope, specifically heptad repeat A, which is common in viral fusion proteins, including all the four viral fusion proteins discussed in this paper. We have provided an additional supplemental figure showing this HRA site on all four structures (Fig. S4). For RSV, this site is known as antigenic site V, and is known to be a highly potent site for**

neutralization, as mAb binding can impair rearrangement to the postfusion conformation. HMPV contains a highly similar HRA region. For HPIV1, we used Alphafold to generate a structure which we superimposed onto one HPIV3 protomer. We have added a discussion of these results (Page 3 lines 52-54, page 4 lines 1-2, page 5 lines 1-2, and page 10 lines 32-35)."

2. The authors also performed in vivo experiments to demonstrate the protection efficacy of these antibodies. While the data did suggest that these antibodies could offer protection, the choice of hamster model might not be ideal for certain pathogens. In addition, the lack of dose titration and lack of benchmark antibodies in the in vivo studies prevents comparison of these antibodies to others previously described in the field.
 - a. *The reviewer brings up an excellent point about the limitations of evaluating protection of human viruses in any small animal model. Upper and lower respiratory tract replication of all the viruses in this study can be demonstrated in both hamsters and cotton rats. We chose the hamster model, because hamsters have been used extensively to evaluate vaccine candidates to parainfluenza viruses, RSV, and HMPV. It is unclear whether cotton rats vs. hamsters are a superior model for RSV, HMPV, or parainfluenza viruses. For HPIV1, the hamster model has been used to evaluate replication of wild-type virus and live attenuated vaccine candidates (PMID 18614629). For HMPV, the hamster model has been used extensively to evaluate HMPV pathogenesis and vaccine efficacy (PMID 18559924, 26063237, 17872522, 18519001, 15194769). The paper referenced by the reviewer (below in point #10) from Wyde, et al. (PMID 15781133) does have a one log higher titer of 4.4 in the lungs of cotton rats on day 4, although the titer declined to 3.3 on day 7. We measured our titers in hamsters on day 5, so timing may have contributed to the difference in viral titer. Another potential reason for the difference in lung titers is that the inoculum used in the paper by Wyde, et al. was approximately 9-fold higher and measured by TCID₅₀ in contrast to our method of counting plaques. For RSV, comparable viral titers have been reported in the lungs of hamsters and cotton rats (PMID 19394861). We have added a discussion of these references (Page 8 lines 3-10).*
 - b. *We have added the results from dose titration experiments and included palivizumab as a benchmark antibody (Fig. 5c, d). Using these data, we have provided an EC90 for each antibody (Fig. 5e). The 5 mg/kg dose was chosen, because it significantly blocked replication of all viruses and is a clinically relevant dose in humans. We also now include the serum concentration of antibodies after intramuscular injection (Fig. 5b). We have included a discussion of these additional results (Page 8 lines 13-16, page 9 lines 1-10).*
3. One weakness of this paper is that only a few antibodies are described. Of note, only two B cell clones were selected out of 900. It would be more interesting if the authors could profile more PIV antibodies as knowledge in the field is scarce. In the field of RSV, it is known the prefusion specific Abs tend to be more potent; however, this is not true for hMPV as the equivalent antigenic site 0 is shielded with glycan thus not immune dominant. Little is known for PIV F antibodies, therefore, a comprehensive profiling of PIV antibodies targeting multiple epitopes could add a lot more value.
 - a. *We agree with the reviewer about the dearth of knowledge in the field regarding PIV antibodies. We performed a more systematic analysis of antibodies targeting HPIV3 in a previous manuscript (PMID 33876699). Similar to RSV, the prefusion conformation of the HPIV3 F protein elicits higher neutralizing antibody titers compared to the postfusion conformation of the F protein (PMID 30420505). We have added a discussion of these references (Page 2 lines 12-20).*

Specific comments:

4. P1, line 30: this is not accurate, at least Moderna has announced efforts to develop HMPV/PIV vaccine.
 - a. *We thank the reviewer for this correction. We have changed the wording to indicate that vaccines against these viruses are not yet clinically available (Page 1 lines 30-31).*
5. P3, line 9: while prefusion specific antibody might be more potent neutralizer (such as RSV site 0 Abs), there are plenty of examples showing antibodies targeting both preF/postF could also be very potent. Authors should evaluate those B cell clones and might obtain more cross-reactive antibodies that target different epitopes than 3x1.
 - a. *Prefusion F is associated with higher neutralizing titers for RSV (PMID 28111638), HPIV1, and HPIV3 (PMID 30420505) which is why we focused on preF-specific B cells in this study. Even though antibodies targeting the post-fusion conformation of HMPV F also contribute to neutralizing antibody titers (PMID 29142300), it is important to note that HMPV postfusion F does not elicit cross-neutralizing antibodies to RSV (PMID 27611367). Since our study focused on identifying cross-neutralizing antibodies, we deliberately focused on the prefusion conformation. Using this approach, we were able to successfully identify candidates for further development. We have added a discussion of this rationale (Page 2 lines 12-20).*
6. P3, line 32: why would only one Fab bound to the F trimer for a large portion of the particles, given that one Fab exclusively contacts only one F protomer? I assume there is excess of Fab vs. F? Author might want to discuss if binding one Fab to the trimeric F could potentially change its conformation and prevent further binding of additional Fabs.
 - a. *To determine how 3x1 interacts with site X, we used cryo-EM. Many HPIV3 preF trimer particles had less than 3 Fabs bound, despite the Fab being in molar excess of trimer during sample preparation. A single peak was observed and collected for the 3x1:HPIV3 complex by size exclusion chromatography (SEC) during sample preparation (SEC trace now added to Fig. S3a). We obtained a structure of 1 Fab in complex with HPIV3 resolved to 3.62 Å (Fig. S2a and Table S1). We also obtained a structure of 3 Fabs bound to HPIV3; this map was limited to 4.3 Å resolution (Fig. S2a and S3a). We noticed no significant variation between the bound protomer and unbound protomer (RMSD=0.721 over 360 Ca) in the C1 structure. Consequently we used the higher resolution structure for model building. Non-stoichiometric complex formation is relatively common in cryo-EM sample preparation. We show that there was no major conformational change between bound and unbound protomers, and that while a three Fab complex was present, its prevalence was lower than the one Fab variant. We believe this may be due to complex instability with the preF protein, or perhaps dissociation due to strong preferred orientation and damage at the air-water interface. We have added a discussion of these results (Page 3 lines 36-41).*
7. Figure 3d: not clear why D25 is not included in the neutralization assay.
 - a. *We have added neutralization data for D25 (Fig. 3). D25 does have greater neutralizing potency compared to MxR against RSV-A, but D25 does not neutralize HMPV. We have included a discussion of these results (Page 6 lines 15-17).*
8. P7, line 1-11: authors discussed potential structural explanations why MxR could bind/neutralize hMPV better than MPE8 and ADI-19425. However, the structures were compared from complexes with RSV F but not hMPV F, while binding to RSV F are similar among these antibodies. CryoEM structures with hMPV F would be more appropriate to offer these insights.
 - a. *While performing experiments with recombinant fusion protein, we assessed the constructs for their potential to withstand cryo-EM. We settled on HPIV3 and RSV as*

*they had the appropriate stability to undergo freezing. Attempts to visualize the HMPV and HPIV1 proteins using cryo-EM presented significant challenges and would require significant further work to overcome (stabilizing mutations, stability tests, expression optimization, and additional microscope time to test conditions). For HMPV, we believe the sequence similarity (**Fig. S6b**) and existing structures provide the necessary information. We have included a figure showing HMPV F modeled with the antibodies of interest in lieu of an additional structure (**Fig. S12**). Please see **Page 6 lines 32-34 and 45-54** for a discussion of these analyses.*

9. Figure 5: please explain the choice of dose 5 mg/kg. It would be more informative if authors could perform a titration of dose levels and calculate EC90 to establish the required levels of antibody for protection. Also, it is not known what are the concentrations of the antibodies in the serum at day 0, this would allow better comparison among individual animals as well as to literature.
 - a. *Please see our response above to point #2.*
10. P7, line 41: in the PBS control, PIV3 established much higher titers in the lung compared to PIV1, authors should discuss whether hamster is a good model for PIV1 (and similarly for hMPV). At least for hMPV, higher viral titers have been established in cotton rats (Wyde et al., doi:10.1016/j.antiviral.2004.12.009).
 - a. *Please see our response above to point #2.*
11. Figure 6: authors described a co-infection model. Because of the plaque assay used can't differentiate RSV vs. PIV3 viruses, it is unclear whether there is interference between the two viruses, i.e. whether both viruses are replicating well when given together. Authors did show qPCR data for the co-infection model, it would be beneficial to compare this data with copy numbers from animals infected with single virus. Ultimately, it would be better to establish plaque assays that differentiate among viruses.
 - a. *The reviewer brings up a good point. We now include qPCR data of RSV and HPIV3 replication in the lungs of hamsters infected with a single virus vs co-infected with two viruses (**Fig. 6a, b**). We did not observe any interference in viral replication during co-infection in our model in which animals were inoculated with equal amounts of virus simultaneously. This is consistent with clinical data in humans suggesting co-infections between other viruses and RSV do not impact RSV titers (PMID 21668660). We have added a discussion of these results (**Page 9 lines 20-24**).*
12. P8, line 53: model PIV1 F structure with 3x1 would be beneficial to understand the conserved mechanism of cross-neutralization. since PIV3 also shares structural homology to RSV/hMPV, authors could discuss if the 3x1 epitope is unique to PIV1/3 or has been previously described for RSV/hMPV antibodies.
 - a. *The 3x1 mAb targets the heptad repeat epitope, specifically heptad repeat A, which is common in viral fusion proteins, including all the four viral fusion proteins discussed in this paper. We have provided an additional supplemental figure showing this HRA site on all four structures (**Fig. S4**). For RSV, this site is known as antigenic site V, and is known to be a highly potent site for neutralization, as mAb binding can impair rearrangement to the postfusion conformation. HMPV contains a highly similar HRA region. For HPIV1, we used Alphafold to generate a structure which we superimposed onto one HPIV3 protomer. We have added a discussion of these results (**Page 3 lines 52-54, page 4 lines 1-2, page 5 lines 1-2, and page 10 lines 32-35**). We did attempt cryo-EM with HMPV and HPIV1 F and found that their instability significantly compromised our ability to generate consistent, stable complexes. This is consistent with previous attempts to*

produce the HPIV1 F protein in the prefusion conformation, and the reason for the bait-and-switch isolation methodology described in this paper.

13. P9, line 5: again, the structures of MxR and MPE8 were compared from complexes with RSV F but not hMPV F, while binding to RSV F are similar among these antibodies. CryoEM structures with hMPV F would be more appropriate to offer these insights.

a. Please see our response above to point #8.

14. P9, line 15: authors should note that other monoclonal antibodies are also in clinical development and has equal potency against RSV A and B (e.g. Tang et al., <https://doi.org/10.1038/s41467-019-12137-1>).

a. We thank the reviewer for noting this reference and have added mention of it in our discussion (Page 10 lines 50-51).

Reviewer #2: The stated goal of these studies was to develop a reagent that would protect against infections of four respiratory pathogens, HPIV1, HPIV3, RSV, and HMPV for, ostensibly, use in immunocompromised populations. The approach was to develop a mAb that cross reacted with the closely related HPIV1 and HPIV3 and a mAb that cross reacted with the closely related RSV and HMPV. The goal was to determine if a cocktail of these two mAb would protect against the four pathogens. While cross reactive mAbs to RSV and HMPV have been reported, none have been developed for the HPIV1 and HPIV3, to my knowledge. The goal of combining such mAbs for protection from four pathogens is moderately novel.

These studies are quite comprehensive and detailed, proceeding from isolation and characterization of the mAb to assays of their protection of animals from challenge.

The authors describe in considerable detail their approaches for finding such antibodies. One protocol was used for developing the human parainfluenza antibody, named 3x1. This task was made more difficult by the failure to isolate a pre-fusion HPIV1 F protein and the authors have devised a clever way around this difficulty. They termed this a "bait-and-switch" strategy. Other investigators might find this approach useful for developing other cross-reacting antibodies. The approach for developing the cross reacting mAb for RSV and HMPV, named MxR, was somewhat more straight forward. The authors cloned and expressed these two antibodies from plasmids and used the purified antibodies to characterize the binding and affinities of the antibodies to three of the target F proteins as well as their neutralizing activities, separately, against each of the four viruses. The authors also characterized the cryo-EM structures of the Fab of mAb 3x1 bound to HPIV3 F protein and cryo-EM structures of the Fab of mAb MxR bound to RSV F protein. Where possible, the authors have compared their results with those of previously isolated mAb to some of the pathogens. While analyses of cryo-EM structures are not complete, missing structures of mAb complexes with HPIV 1 and HMPV F proteins, they do offer a preliminary view of possible mechanisms of neutralization. The authors then proceeded to show that the mAb protected hamsters separately challenged with RSV, HMPV, HPIV1, and HPIV3. Protection from replication of each virus in the lungs was generally good but protection from replication nasal turbinates was negligible. The authors did proceed to test the protection from co-infection with RSV and HPIV3 by a cocktail of MxR and 3x1 showing that upon virus challenge total plaques in lungs (there was no identification of virus in plaques which is quite possible) were considerably reduced, but titers in nasal washes were not reduced significantly. Assay of specific virus load by virus specific qPCR showed less protection. These animal studies are somewhat incomplete and can only be viewed as a preliminary indication of the potential efficacy of the antibody cocktail.

Overall, while these studies are comprehensive, there are some issues that the authors should consider.

1. Some of the figures could be of better quality. Some of the details in the cryo-EM structures described in the text are difficult to impossible to see in the figures. Notably figure 2, panels c-e, and figure 4, panels b-d, should be improved. Colors and printing are hard to see.
 - a. *We are grateful to the reviewer for their overall highly positive comments. High-resolution images have been included with the resubmission. The red/red coloration of MxR has been changed to help with clarity. The higher resolution images show residue detail and lettering sufficient to be read in both online and print format.*
2. In the animal experiments, there was no indication of the sex of the hamsters and no consideration of differences in responses in male or female animals. This issue is receiving increased emphasis in study sections as well as journal reviews since there are often differences in responses to immunization and infection in males and females.
 - a. *An association between sex and clinical outcomes has not been observed in adults with RSV, HMPV, HPIV3, or HPIV1 (PMID 32449300, 23876395). Female hamsters tend to be aggressive and require housing in individual cages. Therefore, due to previous evidence in the literature that sex was not a relevant biological variable and because using female hamsters would have required housing hundreds of hamsters individually, we chose to focus on male animals. We have included a discussion of this (Page 14 lines 16-17).*
3. The animal studies are difficult to interpret since the replication of two different viruses in co-infections has not been characterized. The authors may want to comment on the limitations of these studies here.
 - a. *The reviewer brings up a good point. As discussed in our response to reviewer #1 point #11, “We now include qPCR data of RSV and HPIV3 replication in the lungs of hamsters infected with a single virus vs co-infected with two viruses (Fig. 6a, b). We did not observe any interference in viral replication during co-infection in our model in which animals were inoculated with equal amounts of virus simultaneously. This is consistent with clinical data in humans suggesting co-infections between other viruses and RSV do not impact RSV titers (PMID 21668660). We have added a discussion of these results (Page 9 lines 20-24).”*
4. Did the authors compare protection from challenge in co-infections and single infections with a cocktail of four different mAb each targeting one of the four pathogens with protection using their 3x1 and MxR cocktail? The authors should comment on why their cocktail is better than a 4x cocktail of separate mAb to all four pathogens or a 2x cocktail of an mAb to RSV and a mAb to HPIV3.
 - a. *The reviewer makes a good point. We were unable to administer a cocktail of four antibodies because neutralizing antibodies to HPIV1 (other than the 3x1 mAb we isolated in the present study) were not readily available. The rationale for using fewer antibodies in a mixture is that a higher dose of each antibody can be safely administered in the clinical setting. Mixing more antibodies limits the maximum dose achievable for each antibody, which could limit efficacy. With regards to a 2x cocktail, a cocktail that includes coverage of HMPV and HPIV1 provides greater benefit by increasing breadth of protection for prophylaxis. This is clinically relevant, because all four viruses together account for the majority of serious respiratory viral infections in hematopoietic stem cell transplant recipients. We have included a discussion of these points (Page 9 lines 11-15).*
5. The authors should comment on the potential of mAb resistant mutants and protection from two viruses.
 - a. *As discussed in our response to reviewer #1 point #1, “We also agree that escape mutations are important to consider for further clinical development. We have included a discussion of escape mutations (Page 10 lines 53-54), but this analysis is out of the scope of this*

manuscript.”

6. The authors should comment on why cryo-EMs of mAb with HPIV1 and HMPV were not done.
 - a. *As discussed in our response to reviewer #1 points #8 and #12:*
 - i. *“While performing experiments with recombinant fusion protein, we assessed the constructs for their potential to withstand cryo-EM. We settled on HPIV3 and RSV as they had the appropriate stability to undergo freezing. Attempts to visualize the HMPV and HPIV1 proteins using cryo-EM presented significant challenges and would require significant further work to overcome (stabilizing mutations, stability tests, expression optimization, and additional microscope time to test conditions). For HMPV, we believe the sequence similarity (Fig. S6b) and existing structures provide the necessary information. We have included a figure showing HMPV F modeled with the antibodies of interest in lieu of an additional structure (Fig. S12). Please see Page 6 lines 32-34 and 45-54 for a discussion of these analyses.”*
 - ii. *“The 3x1mAb targets the heptad repeat epitope, specifically heptad repeat A, which is common in viral fusion proteins, including all the four viral fusion proteins discussed in this paper. We have provided an additional supplemental figure showing this HRA site on all four structures (Fig. S4). For RSV, this site is known as antigenic site V, and is known to be a highly potent site for neutralization, as mAb binding can impair rearrangement to the postfusion conformation. HMPV contains a highly similar HRA region. For HPIV1, we used AlphaFold to generate a structure which we superimposed onto one HPIV3 protomer. We have added a discussion of these results (Page 3 lines 52-54, page 4 lines 1-2, page 5 lines 1-2, and page 10 lines 32-35). We did attempt cryo-EM with HMPV and HPIV1 F and found that their instability significantly compromised our ability to generate consistent, stable complexes. This is consistent with previous attempts to produce the HPIV1 F protein in the prefusion conformation, and the reason for the bait-and-switch isolation methodology described in this paper.”*

Reviewer #3: In this manuscript the authors present the discovery and characterization of two cross-reactive antibodies, which they named 3x1 and MxR. The 3x1 mAb has high neutralization potency against both human parainfluenza virus-1 (HPIV1) and HPIV3 and demonstrates tight binding to the prefusion conformation of the HPIV3 F protein (preF). Similarly, the MxR antibody is highly potent against both respiratory syncytial virus (RSV) and human metapneumovirus (HMPV) and binds the preF proteins of RSV and HMPV with nanomolar or higher affinity. The authors further used cryo-electron microscopy (cryo-EM) to solve structures of Fab 3x1 complexed with preF of HPIV3 and of Fab MxR complexed with PreF of RSV, providing structural basis for tight binding and cross-neutralization. In vivo experiments, the two mAbs were highly efficient in protecting golden Syrian hamsters against the corresponding infections. Importantly, a cocktail of 3x1 and MxR provided protection against HPIV1, HPIV3, RSV and HMPV simultaneously. This makes the combination of 3x1 and MxR a promising candidate for further development, with potential clinical utility in protecting immunocompromised individuals from the four respiratory viruses, which cumulatively account for the majority of lower respiratory tract infections in vulnerable populations. These findings are novel and highly important. I believe this manuscript deserves publication after the issues outlined below are resolved. My comments and questions are mostly focused on validation and presentation of the cryo-EM structures.

Major issues and questions related to structural analysis

1. Some essential cryo-EM validation information is absent from the manuscript. Please include the following in the Supplementary Figures:

- a. The FSC plot for each map as generated by cryoSPARC, with all the curves present. Currently only the FSC plot of the 3-Fab 3x1 map is included (Fig. S2a), but it appears to be cropped on the right side (unless the data has been binned). Presenting only local resolution “heat maps” is not sufficient.
 - a. *We are grateful to the reviewer for the careful and thorough reading of the manuscript. GSFSC plots have been added for both deposited structures (Fig. S2). The data for the 3-Fab 3x1 map was binned to 2.0577 Å/px, and the GSFSC curve is uncropped (Fig. S3).*
 - b. Cryo-EM density overlaid with the atomic model for the CDRs that are in contact with preF (for both high-resolution maps).
 - a. *We have added the CDR density (Fig. S5 and S10).*
 - c. Estimates of map-model correlations in some form. Map-model FSC curves (for example, generated with phenix.mtriage) would be helpful.
 - a. *Phenix Model:Map FSC plots have been added (Fig. S2).*
 - d. A representative micrograph for each complex.
 - a. *Representative micrographs have been added (Fig. S2).*
2. The 3x1 mAb has picomolar or even higher affinity to HPIV3 PreF protein (page 3, line 21). Nonetheless (and unlike in the case of the complex between Fab MxR and RSV PreF), size exclusion chromatography apparently isolated mostly non-stoichiometric complexes, with the complex between one Fab and one PreF trimer being the main species according to the cryo-EM results. I have two suggestions:
- a. To determine whether binding of one 3x1 Fab hinders subsequent Fab binding by altering the global conformation of the PreF protein, the authors should superpose and compare the protomers of HPIV3 PreF in the structure of the complex. Are there any significant differences between the protomers’ structures? Furthermore, does the PreF trimer in this complex deviate noticeably from the expected C3 symmetry?
 - a. *Please see our response to reviewer #1 point #6:*
 - i. *“To determine how 3x1 interacts with site X, we used cryo-EM. Many HPIV3 preF trimer particles had less than 3 Fabs bound, despite the Fab being in molar excess of trimer during sample preparation. A single peak was observed and collected for the 3x1:HPIV3 complex by size exclusion chromatography during sample preparation (Fig. S3a). We obtained a structure of 1 Fab in complex with HPIV3 resolved to 3.62 Å (Fig. S2a and Table S1). We also obtained a structure of 3 Fabs bound to HPIV3; this map was limited to 4.3 Å resolution (Fig. S2a and S3a). We noticed no significant variation between the bound protomer and unbound protomer (RMSD=0.721 over 360 Cα) in the C1 structure. Consequently we used the higher resolution structure for model building. Non-stoichiometric complex formation is relatively common in cryo-EM sample preparation. We show that there was no major conformational change between bound and unbound protomers, and that while a three Fab complex was present, its prevalence was lower than the one Fab variant. We believe this may be due to complex instability with the preF protein, or perhaps dissociation due to strong preferred orientation and damage at the air-water interface. We have added a discussion of these results (Page 3 lines 36-41).”*
 - b. It would help if the authors measured the affinity of 3x1 Fab and compared it to the affinity of mAb 3x1. It is possible that both Fabs of the antibody need to interact with preF simultaneously for tight binding. If so, the mechanism of neutralization could be more complicated than simply via blocking the furin cleavage site as suggested in the Discussion.

- a. *The reviewer makes an excellent point. We have included data on binding kinetics for the 3x1 Fab (Fig. 1d). 3x1 Fab indeed binds with lower affinity compared to 3x1 IgG. We have added a discussion of these results (Page 3 lines 21-24).*
 - b. *Please also see response to reviewer #1 point #1 regarding a discussion of the mechanism of neutralization in which we make the following points:*
 - i. *“We have included data from a fusion inhibition assay. Using fluorescence microscopy, we show that 3x1 blocks cell-to-cell spread and syncytia formation by HPIV3 in vitro (Fig. S1). We considered performing a furin cleavage assay, but the binding assays and structural data indicate that 3x1 binds to the mature prefusion conformation of the F protein following furin cleavage (Fig. S7). Therefore, 3x1 would not be expected to inhibit furin cleavage. We have added a discussion of these methods and results (Page 3 lines 26-27).”*
 - ii. *“The 3x1 mAb targets the heptad repeat epitope, specifically heptad repeat A, which is common in viral fusion proteins, including all the four viral fusion proteins discussed in this paper. We have provided an additional supplemental figure showing this HRA site on all four structures (Fig. S4). For RSV, this site is known as antigenic site V, and is known to be a highly potent site for neutralization, as mAb binding can impair rearrangement to the post fusion conformation. HMPV contains a highly similar HRA region. For HPIV1, we used Alphafold to generate a structure which we superimposed onto one HPIV3 protomer. We have added a discussion of these results (Page 3 lines 52-54, page 4 lines 1-2, page 5 lines 1-2, and page 10 lines 32-35).”*
3. The route that the authors took to arrive at the final cryo-EM map of the 3x1 complex with one Fab bound is unusual. Since, according to the Methods and Fig. S1a, the ab initio model generation produced one clear “winner” – with one Fab bound to one trimer – it made sense to refine the population of particles showing one Fab against this model. Why did the authors then choose to impose C3 symmetry on clearly asymmetrical particles and then relax this symmetry, followed by the removal of the particle duplicates that were produced in the process? It seems that doing this should be equivalent to doing nothing and should not change the map. Although the authors state in the Methods (page 12, line 14) that this approach produced minor improvements in the map, the resolution stayed the same (3.6 Å). What was the nature of these improvements and are they quantifiable? I would like to note that refinements of cryo-EM maps in complicated cases often turn into a “kitchen sink,” and the path taken by the authors is not by itself disqualifying.
- a. *We agree that this was a somewhat unusual refinement path and that a simpler path may have produced an identical map. During subsequent refinement of the original 1-Fab ab-initio model, we noticed some partial Fab density at the binding site of the adjacent protomers. 2D and 3D classing were unable to isolate sub-groups of 2 or 3-Fab particles. We assumed these were “misaligned” 1 Fab particles due to the size/signal difference between the larger protomer (180 kDa) and the much smaller Fab (48 kDa). The symmetry refinement and relax protocol was borrowed from work with a complex with similar symmetry breaking features (overall C1 with local CX symmetry) that worked to align the symmetry breaking feature. While we didn’t see any improvement of overall resolution following this pipeline in 3x1:HPIV3, we did notice a reduction in the partial Fab density at adjacent protomers. In short, while we may not be able to quantify the map improvements in GSFSC plots, we do believe that this pipeline was appropriate, and as mentioned, at worst would produce an unchanged map.*

4. On page 4, lines 2-4 of the Results the authors describe hydrogen bonds within Fab 3x1 and between its residue Arg91 of the light chain and Asp143 of HPIV3 PreF. Is the local resolution sufficient to support this? Can the authors provide close-up view(s) with cryo-EM density or modify their interpretation?
 - a. *We found that at the binding site, local resolution was greater than the overall map resolution, and there is density to support the location of 3x1 Tyr31 LC, Arg91 LC, Leu100F HC, and HPIV3 preF Asp143. As could be expected with multiple binding partners, there is significant density for the location of the Arg91 LC sidechain, which we understand is usually not the case for less coordinated sidechains at lower resolution. Our new supplemental figure (Fig. S5) includes this map density.*

Minor issues and questions related to structural analysis

5. According to Supplementary Table 1, the defocus range for the 3x1 dataset was -0.5 to -3.5 microns, but in the provided validation report it is -1 to -3 microns.
 - a. *We thank the reviewer for catching this inconsistency. This was a typo from the MxR:RSV data and has been corrected.*
6. According to the Methods (page 11, line 59), the electron exposure was 50 e-/Å², but it is different in the Supplementary Table 1 and the validation reports (~41 e-/Å²).
 - a. *The electron dose was 50 e-/Å², and the values in **Supplementary Table 1** and validation reports have been corrected.*
7. In the Methods the authors use the phrase “with a GSFSC resolution significantly greater than Nyquist at 2.0577 Å/px” (page 12, line 3). Perhaps they meant that the Fourier shell correlation value did not drop to 0 at the Nyquist frequency (which for the binned data should be ~4.4 Å - twice the pixel size; also it is not measured in Å/px)? Strictly speaking, exceeding the Nyquist limit violates a fundamental principle of cryo-EM and beyond.
 - a. *We thank the reviewer for catching this error in terminology. We have reworded this section for accuracy (**Page 13 lines 50-62**). The specific section now reads, "Following import, patch motion correction (micrographs binned to 1.02885 Å/px), and contrast transfer function (CTF) estimate, micrographs were curated for < 4 Å CTF fit. Blob picker was used to select ~50,00 particles, which underwent 2D classification to produce templates for template-based picking. These picks were inspected, curated, and extracted (192 pixel box size at 2.0577 Å/px) with 1.49 million particles for MxR:RSV and 2.02 million particles for 3x1:HPIV3. Following two rounds of 2D classification, 100 classes each, we were left with 377,982 MxR:RSV particles showing three Fabs bound. A single ab-initio model was generated and refined. Particles were re-extracted at 1.02885 Å/px and subject to local CTF refinement and non-uniform refinement with C3 symmetry. Following this, particles were further curated and re-extracted using Local Motion Correction, producing 354,958 particles. Non-uniform refinement with a custom mask cropping out the CHI region and GCN4 domain produced a sharpened map (GSFSC = 0.143) of 2.41 Å resolution using C1 symmetry, and 2.24 Å using C3 symmetry."*
8. I suggest that the authors carefully revise and polish the cryo-EM Methods. Some issues include:
 - a. Omission of essential references (cryoSPARC, DeepEMhancer, ChimeraX, ISOLDE, Pymol, PDBePISA, ...).
 - i. *We thank the reviewer for catching these mistakes and have now carefully revised and polished the cryo-EM methods. We have now included these references.*
 - b. What is “COSMIC computer” (page 12, line 18)? The reference is to a review about RSV vaccine development.

- i. *We have added the correct citations (Page 14 line 6). We agree that this was a little confusing as the name of the computer is “COSMIC²”.*
 - c. Colloquial language not appropriate for a manuscript (“Krios 300kV Cryo-EM scope with a K3 camera”) - A Thermo Scientific Titan Krios electron microscope (rather than “scope”) operated at 300 kV; K3 is manufactured by Gatan, etc.
 - i. *This wording has been corrected (Page 13 lines 50-51).*
 - d. Awkward phrases and sentences:
 - i. “Grids were collected on...” – Datasets were collected on...?
 - ii. “with a super resolution of 0.514425 Å/px” – in the super-resolution mode with a pixel size of...?
 - iii. “The template generation for 3x1:HPIV3 produced a mix of 1 Fab, 2 Fabs, and 3 Fabs...” – produced classes with one, two, and three Fabs bound?
 - iv. “Grids were also glow discharged prior to the start of freezing”
 - v. “A small subset of particles (~50,000) was picked, underwent 2D classification, and classes selected for the template-based picking job.”
 - vi. ...and so on
 - a. *We have reviewed the methods and all above comments have been addressed.*
9. Both EMD validation reports alert about a large discrepancy between author provided resolution and author provided FSC curve (3.62 Å vs 3.98 Å for 3x1; 2.24 Å vs 2.81 Å for MxR). Did the authors deposit the FSC curves obtained without masking? This is not a problem for this manuscript (provided that all the FSC plots will be added as suggested above), but maybe the authors will have a chance to re-upload the FSC plots that support the stated resolutions before finalizing the depositions.
- a. *Following the PDB Dep instructions, we had uploaded FSC.xml curves generated from the unmasked half-maps, which report slightly lower resolutions. We also deposited the sharp maps which are of the higher resolutions listed. This was a misunderstanding of which maps needed to be deposited under what listings. This has been corrected in the most recent deposition reports to report the sharp map FSC.xml.*
10. The resolution of the MxR complex is impressive (2.2 Å). Did the authors see density for any water molecules? There are no water molecules in the model according to the table.
- a. *Water molecules have been added to the structure. Due to the significant number of water molecules, the main figures do not include these, and the text has been updated to note this (Page 6 lines 34-35). An additional supplemental figure (Fig. S11) has been generated to show the position of water molecules within the binding site.*
11. The authors state that they did not build a molecular model for the symmetrical 3x1 map, but it is shown in Fig. S2, panels b-d. Was this the one-Fab model docked into the three-Fab map? This should be stated in the legend.
- a. *We indeed did not build a molecular model for the symmetrical 3x1 map. The original figure was comprised of two different models. Panels b-d showed the 1-Fab model and structure; those panels had no relation with the 3-Fab structure shown in panel a. To avoid ambiguity, the supplemental figures have been rearranged and split into two separate figures so that the 3-Fab map structure is shown in Fig. S3a and the 1-Fab model and structure are shown in Fig. S4.*

Other minor issues and questions

12. Fig. 1c: the red bar should be described in the legend.

- a. The red bar indicates data from the well that contained the 3x1-producing B cell. This has been added to the legend for **Fig. 1**.*
13. Fig. 2b, bottom panel: PreF protein colors are not defined. The legend only mentions shades of grey.
 - a. This has been corrected and the colors are now defined.*
14. Fig 2e: the rotation angle is 110 degrees, but it is 100 in the legend.
 - a. The legend has been corrected to 110 degrees.*
15. Fig. 4a: “dark red” and “light red” are difficult to distinguish; please consider a different combination.
 - a. The colors have been modified so they are more distinguishable.*
16. Fig. 4a: where is the II' label? Both labels seem to read just II.
 - a. The label has been corrected to show DII'.*
17. For consistency with Fig. 2, rotation angles should be indicated above panels 4b-d
 - a. This has been corrected and the rotation angles are now shown.*
18. Figure S4 is not mentioned in the manuscript.
 - a. The original Fig. S4 (now **Fig. S7**) is now referenced in the manuscript (**Page 5 lines 9-13**).*
19. A section title states “Identification of a novel cross-neutralizing antibody against multiple parainfluenza viruses” (page 3). Since there are only two such viruses, the title needs to be modified accordingly.
 - a. The section title has been changed to “Identification of a novel HPIV3 and HPIV1 cross-neutralizing antibody” (**Page 3 line 2**).*
20. On page 4, line 4 there is a reference to Fig. S2e. There is no panel e in Fig. S2.
 - a. This has been corrected to refer to **Fig. S5a-c and S6a** (**Page 5 line 9**).*

Reviewers' Comments:

Reviewer #1:

Remarks to the Author:

The authors have addressed most of my concerns through additional data and discussion of limitations and caveats. The study will be a useful additional resource for those in the field interested in understanding immune responses to these respiratory viruses.

Reviewer #2:

Remarks to the Author:

This manuscript describes the isolation of a mAb that cross reacts with HPIV 1 and 3 and another mAb that cross reacts with RSV and HMPV. The goal was to generate two antibodies that, if mixed, could protect against the four respiratory viruses. The methods described are novel and potentially useful to other investigators. The results are promising for the development of a therapeutic for protection of immunocompromised populations. The results are also interesting for considerations of the evolutionary relationships between fusion proteins of respiratory viruses.

This is a submission of a revised manuscript in response to three reviewers all of whom had issues with data analysis and missing experiments. I have carefully read the authors' responses to all three reviews and am satisfied that the authors have responded appropriately to each of my comments and many of those of the other reviewers. However, I do not have experience with cryo-EM. Thus, I cannot comment on how well the authors responded to issues, raised by the other reviewers, concerning generation and analysis of the cryo-EM structures.

Reviewer #3:

Remarks to the Author:

My comments have been addressed satisfactorily. I suggest that the authors make a few small changes in the final version:

1. Add scale bars to the representative micrographs and 2D class average images in Supplementary Figure 2.
2. Enlarge the FSC plots in Supplementary Figure 2. The numbers are very difficult to read.
3. In Supplementary Figure 5 label the residues mentioned in the legend.

We would like to thank the editors and reviewers for the second review and are grateful for the opportunity to respond. The manuscript has been revised and the point-by-point response is found below. Reviewer comments are displayed in regular text, and our responses are in italics.

Reviewer #1 (Remarks to the Author):

The authors have addressed most of my concerns through additional data and discussion of limitations and caveats. The study will be a useful additional resource for those in the field interested in understanding immune responses to these respiratory viruses.

- *We are grateful to the reviewer for their comments and the overall careful reading of our revised manuscript*
-

Reviewer #2 (Remarks to the Author):

This manuscript describes the isolation of a mAb that cross reacts with HPIV 1 and 3 and another mAb that cross reacts with RSV and HMPV. The goal was to generate two antibodies that, if mixed, could protect against the four respiratory viruses. The methods described are novel and potentially useful to other investigators. The results are promising for the development of a therapeutic for protection of immunocompromised populations. The results are also interesting for considerations of the evolutionary relationships between fusion proteins of respiratory viruses.

This is a submission of a revised manuscript in response to three reviewers all of whom had issues with data analysis and missing experiments. I have carefully read the authors' responses to all three reviews and am satisfied that the authors have responded appropriately to each of my comments and many of those of the other reviewers. However, I do not have experience with cryo-EM. Thus, I cannot comment on how well the authors responded to issues, raised by the other reviewers, concerning generation and analysis of the cryo-EM structures.

- *We are grateful to the reviewer for their thoughtful second review of our revised manuscript.*
-

Reviewer #3 (Remarks to the Author):

My comments have been addressed satisfactorily. I suggest that the authors make a few small changes in the final version:

1. Add scale bars to the representative micrographs and 2D class average images in Supplementary Figure 2
 - a. *We thank the reviewer for their comments and feedback on the revised manuscript. We have added scale bars to **Supplementary Figure 2**.*
2. Enlarge the FSC plots in Supplementary Figure 2. The numbers are very difficult to read.
 - a. *We have enlarged the FSC plots in **Supplementary Figure 2** to improve readability.*
3. In Supplementary Figure 5 label the residues mentioned in the legend.
 - a. *We have labeled the residues in **Supplementary Figure 5** that are mentioned in the legend.*